# ORTHORF: EXPLORING ORTHOGONALITY IN OBJECT-CENTRIC REPRESENTATIONS

**Despoina Touska**[1,†], **Bastiaan Onne Fagginger Auer**[2], **Alexandru Onose**[2],
**Tejaswi Kasarla**[1], **Luis Armando Pérez Rey**[2], **Maximilian Lipp**[3], **Lyubov Amitonova**[3]
**Martin R. Oswald**[1], **Pascal Cerfontaine**[4,‡]

[1]Faculty of Science, University of Amsterdam, Amsterdam, the Netherlands
[2]ASML Netherlands B.V., Veldhoven, the Netherlands
[3]Advanced Research Center for Nanolithography, Amsterdam, the Netherlands
[4]Faculty of Information, Media and Electrical Engineering, TH Köln, Cologne, Germany
Correspondence: `d3py.tousk@gmail.com`, `pascal.cerfontaine@th-koeln.de`

## ABSTRACT

Neural synchrony is hypothesized to help the brain organize visual scenes into structured multi-object representations. In machine learning, synchrony-based models analogously learn object-centric representations by storing binding in the phase of complex-valued features. Rotating Features (RF) instantiate this idea with vector-valued activations, encoding object presence in magnitudes and affiliation in orientations. We propose Orthogonal Rotating Features (OrthoRF), which enforces orthogonality in RF's orientation space via an inner-product loss and architectural modifications. This yields sharper phase alignment and more reliable grouping. In evaluations of unsupervised object discovery, including settings with overlapping objects, noise, and out-of-distribution tests, OrthoRF matches or outperforms current models while producing more interpretable representations, and it eliminates the post-hoc clustering required by many synchrony-based approaches. Unlike current models, OrthoRF also recovers occluded object parts, indicating stronger grouping under occlusion. Overall, orthogonality emerges as a simple, effective inductive bias for synchrony-based object-centric learning.

## 1 INTRODUCTION

Decomposing scenes into constituent parts is a long-standing strategy in computer vision. Classical approaches factorized images into surfaces and objects with properties like reflectance, albedo, and geometry (Coakley, 2003). Deep learning has renewed this effort by learning structured representations (Zhang et al., 2013; Dittadi, 2023). In practice, this is often operationalized as Object-Centric Learning (OCL) (Greff et al., 2020), where models discover modular, compositional object representations that support generalization and relational reasoning on many downstream visual tasks (Ding et al., 2021; Bapst et al., 2019; Mandikal & Grauman, 2021).

At its core, much like human perception (Spelke, 1990), OCL addresses the binding problem (Roskies, 1999): flexibly integrating features (such as color, shape, texture) into a unified perception. This view aligns with cognitive and neuroscientific accounts that posit neural synchrony (Singer, 2007) as a key mechanism, whereby temporally synchronized oscillations bind distributed information into coherent objects (see Fig 1).

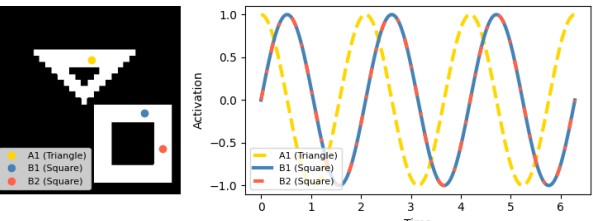

Figure 1: Binding-by-synchrony. Local excitation, inhibition partitions neurons into phase-based groups. Same-phase firing (e.g., B1–2) encodes the same object; out-of-phase firing (e.g., A1 vs. B1-2) encodes different objects.

---

† Research done during an internship at ASML Netherlands B.V, Veldhoven, the Netherlands. ‡Research partially done at ASML Netherlands B.V, Veldhoven, the Netherlands.

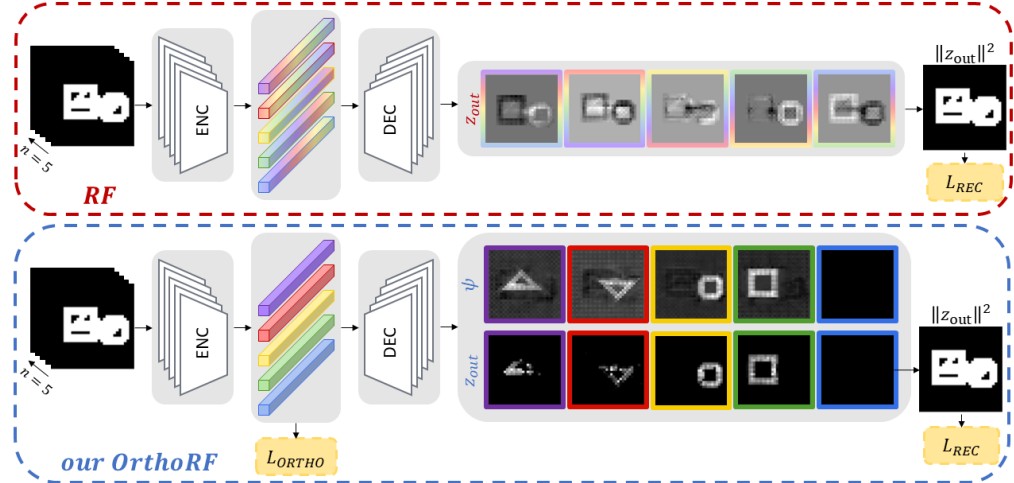

Figure 2: RF vs. our OrthoRF. In RF, object content is distributed across orientation space (seen as multicolored bars in encoder activations and mixed final outputs). OrthoRF enforces orthogonal orientation axes, routing each object to a distinct latent, yielding cleaner separation and improved handling of overlap regions.

A dominant OCL design to binding uses a collection of discrete latent vectors, named "slots" (Locatello et al., 2020), each dedicated to the features of a single object. As slot-based methods evolve, radically new ideas are emerging in OCL, some of which are inspired by synchrony (Mozer et al., 1991; Reichert & Serre, 2013). In this less-explored paradigm, binding is expressed via the relative phases of complex-valued neural activations, with phase-space distances serving as an implicit relational metric between object instances. Recently, synchrony-based models (Löwe et al., 2022; Gopalakrishnan et al., 2024; Miyato et al., 2024) have achieved unsupervised object discovery on synthetic and more naturalistic scenes, surpassing prior supervised approaches (Mozer et al., 1991).

Current state-of-the-art synchrony-based models, such as Complex-valued Autoencoders (CAE) (Löwe et al., 2022), use real-valued weights to process complex-valued activations, by sharing weights across the real and imaginary parts. In these models, magnitudes encode feature presence, whereas phases encode object affiliation. During training, they leverage the natural constructive/destructive interference through the addition of complex activations in every layer, promoting phase alignment for features of the same object and phase separation for different objects. A related work, named Rotating Features (RF) (Löwe et al., 2024a), replaces complex-valued with vector-valued (i.e., $n$-dimensional) activations that rotate on a hypersphere, extending representational capacity beyond 2D complex planes. To (de)synchronize phases for object representation, these models use an additional inductive bias via a gating mechanism (Reichert & Serre, 2013) that strengthens interactions among similarly oriented features and weakens those among dissimilarly oriented ones. However, this mechanism can be hard to interpret. To improve interpretability, Löwe et al. (2024b) introduced cosine binding, a more transparent alternative that is based on cosine similarity between activations. This, however, entails substantial memory overhead, since it requires computing and storing many similarities between inputs and intermediate outputs.

Synchrony-based models have important limitations. Unlike slot-based methods, which yield discrete, object-aligned slots (one slot $\approx$ one object), state-of-the-art synchrony-based models (Löwe et al., 2022; 2024a; Stanić et al., 2023; Miyato et al., 2024; Gopalakrishnan et al., 2024) produce distributed representations. While this can be more flexible, it makes the output hard to use without extra machinery. In practice, they require post-hoc clustering in phase space to recover objects, grouping features by phase. This distributed coding also degrades performance in overlap regions: features from occluding objects become uncertain and drift farther from cluster centers (Löwe et al., 2024a), complicating object assignment. As a result, evaluations of these models often exclude overlapping regions, potentially underrepresenting the regimes where robust binding is most needed.

In this paper, we advance synchrony-based models, specifically the RF autoencoder (Löwe et al., 2024a), by improving both interpretability and representational capacity. Motivated by evidence

that orthogonality enhances efficiency and encourages disentanglement (Ranasinghe et al., 2021), we propose Orthogonal Rotating Features (OrthoRF), an RF extension that enforces orthogonal object encoding in the phase space. Our approach has two components: (i) a softmax-based competitive binding that drives $n$-dimensional activations to specialize on distinct input components (objects), and (ii) an inner-product-based orthogonality loss that enforces a $90°$ separation between object representations in phase space. Together, these yield sharper phase alignment and more reliable grouping: features of the same object concentrate along a single vector dimension, producing one-hot–like object encodings. In unsupervised object discovery evaluation, across scenarios with overlapping objects, noise, and out-of-distribution tests, OrthoRF matches or surpasses current methods, eliminates post-hoc clustering, and recovers occluded object parts in intermediate representations (a capability not shown by slot-based or prior synchrony-based models). These results underscore orthogonality as a simple yet powerful inductive bias for synchrony-based object-centric learning.

## 2 BACKGROUND: ROTATING FEATURES

We build upon the RF autoencoder (Löwe et al., 2024a), which replaces scalar features with $n$-dimensional vectors whose magnitudes encode feature presence and whose orientations encode object affiliation. Specifically, a standard feature vector $\mathbf{z} \in \mathbb{R}^d$ is lifted to $\mathbf{z}_{\text{rotating}} \in \mathbb{R}^{n \times d}$, whose per-feature magnitude $\mathbf{m} = \|\mathbf{z}_{\text{rotating}}\|_2 \in \mathbb{R}^d$ (the $\ell_2$-norm over the $n$-dimension) plays the role of a standard neural activation. This lifting applies both to input images (initialized with zeros along the orientation dimension) and to activations at any layer. Given a neural layer $f_{\mathbf{w}}$ with $d_{\text{in}}$ inputs and $d_{\text{out}}$ outputs, an input $\mathbf{z}_{\text{in}} \in \mathbb{R}^{n \times d_{\text{in}}}$ is transformed using a weight matrix $\mathbf{w} \in \mathbb{R}^{d_{\text{in}} \times d_{\text{out}}}$, shared across the $n$ components, and a bias $\mathbf{b} \in \mathbb{R}^{n \times d_{\text{out}}}$, as follows:

$$\boldsymbol{\psi} = f_{\mathbf{w}}(\mathbf{z}_{in}) + \mathbf{b} \ \in \mathbb{R}^{n \times d_{out}}. \tag{1}$$

To ensure that similar oriented features are processed together, RF uses a gating mechanism[0] (Reichert & Serre, 2013). Specifically, it applies the shared weights $\mathbf{w} \in \mathbb{R}^{d_{\text{in}} \times d_{\text{out}}}$ to both the inputs $\mathbf{z}_{\text{in}} \in \mathbb{R}^{n \times d_{\text{in}}}$ and their per-feature magnitudes $\|\mathbf{z}_{\text{in}}\|_2 \in \mathbb{R}^{d_{\text{in}}}$, then combines the results:

$$\boldsymbol{\chi} = f_{\mathbf{w}}(\|\mathbf{z}_{\text{in}}\|_2) \in \mathbb{R}^{d_{\text{out}}}, \quad (2) \qquad \mathbf{m}_{\text{bind}} = 0.5 \cdot \|f_{\mathbf{w}}(\mathbf{z}_{\text{in}})\|_2 + 0.5 \cdot \boldsymbol{\chi} \in \mathbb{R}^{d_{\text{out}}}. \quad (3)$$

The gated magnitude $\mathbf{m}_{\text{bind}}$ is passed through ReLU to enforce non-negativity, and then used to rescale $\boldsymbol{\psi}$, ensuring the output of the layer retains $\boldsymbol{\psi}$'s orientation, as follows:

$$\mathbf{m}_{out} = \text{ReLU}(\text{BatchNorm}(\boldsymbol{m}_{\text{bind}})) \in \mathbb{R}^{d_{\text{out}}}, \quad (4) \quad \mathbf{z}_{\text{out}} = \mathbf{m}_{out} \odot \frac{\boldsymbol{\psi}}{\|\boldsymbol{\psi}\|_2} \in \mathbb{R}^{n \times d_{\text{out}}}. \quad (5)$$

The reconstructed image is obtained by computing the per-pixel magnitude of the final-layer activations, $\|\mathbf{z}_{\text{final}}\|_2 \in \mathbb{R}^{d_{\text{image}}}$ with $d_{\text{image}} = c \times h \times w$ (where $c, h, w$ are channels, height, and width), scaling it with a learnable scalar weight $w' \in \mathbb{R}$ and bias $b' \in \mathbb{R}$, and then applying a sigmoid:

$$\hat{\mathbf{x}} = \text{Sigmoid}(w'\|\mathbf{z}_{\text{final}}\|_2 + b') \ \in \mathbb{R}^{d_{\text{image}}}. \tag{6}$$

During training, an MSE loss is used between the input and reconstructed images, $\mathcal{L}_{\text{REC}} = \text{MSE}(\mathbf{x}, \hat{\mathbf{x}})$. The vector-valued activations add across layers, producing constructive interference for features of the same object and destructive interference for different objects. Because regions of the same object exhibit high pointwise mutual information, destructive interference would hurt reconstruction, so training implicitly encourages within-object alignment and across-object anti-alignment in features. For object discovery, $k$-means is applied to the output $\mathbf{z}_{\text{final}} \in \mathbb{R}^{n \times d_{\text{image}}}$, assigning each pixel to an object-cluster. See Löwe et al. (2024a) for further details on RF.

---

[0]There is an inconsistency between the description in the paper and the implementation in the code regarding the calculation of $\mathbf{m}_{bind}$. While the paper states that $\mathbf{m}_{bind} = 0.5 \cdot \|\boldsymbol{\psi}\|_2 + 0.5 \cdot \boldsymbol{\chi}$, the code utilizes the formula described in Equation 3 for $\mathbf{m}_{bind} \in \mathbb{R}^{d_{out}}$. We used the latter implementation as it performs better.

## 3 METHOD

### 3.1 MOTIVATION

Synchrony-based architectures such as RF (Löwe et al., 2024a) show that these $n$-dimensional features can support object discovery, yet their representations are distributed (see Fig. 2) across dimensions, which in turn demands post-hoc clustering (e.g., $k$-means) to recover objects. This dependence makes the pipeline fragile and less practical; a single object may occupy multiple dimensions, creating redundancy and blurring boundaries, particularly in overlap regions where features drift away from cluster centers (as noted by Löwe et al. (2024a)) and assignments become uncertain. That uncertainty, however, carries informative cues: RF's behavior in overlaps reveals occlusion signals that slot-based OCL methods (Anciukevicius et al., 2020) seldom exploit. We investigate whether an architectural bias can preserve RF's strengths while addressing these drawbacks. Guided by evidence that orthogonality sharpens discrimination and fosters disentanglement (Lezama et al., 2018; Ranasinghe et al., 2021; Sun et al., 2017; Chen et al., 2020; Liu et al., 2018; Wang et al., 2018), we impose orthogonality constraints in the RF orientation space so that each object collapses onto a single component in this $n$-dimensional orientation space, reducing redundancy, removing the need for clustering, and converting overlap-driven uncertainty into a reliable cue for occlusion recovery.

### 3.2 ORTHOGONAL ROTATING FEATURES

In this section, we present the architectural modifications that yield competitive binding, and an orthogonality loss enforcing $90°$ separation among latents in the orientation space. We also highlight key properties that emerge from these modifications. Fig. 2 illustrates the overall OrthoRF architecture.

**Competitive binding in orientation space**   We model object–component assignment as a discrete competition: each object should map to one component in the $n$-dimensional orientation space. Inspired by multi-class classification, where a softmax layer maps logits to a categorical distribution, and by OCL methods such as Slot Attention (SA) (Locatello et al., 2020), we use the same mechanism to induce competition that drives object-oriented specialization across components. In OrthoRF autoencoder, we apply a per-layer softmax over orientation components, yielding winner-take-most assignments and object-wise specialization. To improve stability and prevent component collapse (e.g., all features mapped to one component while others are never used), we apply centering before the softmax, but only to the encoder's output vectors. Empirically, this removes biases that would otherwise let a single component dominate (Caron et al., 2021). Specifically, after Eq. 1, we use the intermediate output $\psi \in \mathbb{R}^{n \times d}$ (rows $i$: orientation components; columns $j$: features). For each feature index $j$, we apply softmax over components $i$ after subtracting the per-feature mean logit to obtain assignment probabilities, as follows:

$$\psi'_{ij} = \frac{\exp\left(\psi_{ij} - \bar{\psi}_j\right)}{\sum_{k=1}^{n} \exp\left(\psi_{kj} - \bar{\psi}_j\right)}, \quad \text{where} \quad \bar{\psi}_j = \frac{1}{n} \sum_{k=1}^{n} \psi_{kj}. \tag{7}$$

The remaining steps follow Section 2.

**Orthogonality regularization**   We enforce orthogonality among latent orientation components at the encoder output, since this stage aggregates global features and offers a lower-dimensional representation, reducing computational cost. To implement this, we use the encoder's output $\mathbf{z} \in \mathbb{R}^{\text{bs} \times n \times z_{\dim}}$, (bs: batch size, $n$: orientation components, $z_{\dim}$: feature dimension). For each sample $i \in \{1, \dots, \text{bs}\}$ and feature $j \in \{1, \dots, z_{\dim}\}$, we center across orientation components:

$$\tilde{z}_{ikj} = z_{ikj} - \bar{z}_{ij}, \qquad \text{where} \qquad \bar{z}_{ij} = \frac{1}{n} \sum_{m=1}^{n} z_{imj}. \tag{8}$$

After centering, we stack the $n$ latent vectors for a sample $i$ as the rows of $\tilde{Z}_i \in \mathbb{R}^{n \times z_{\dim}}$ and define the Gram matrix, as follows:

$$G_i = \tilde{Z}_i \tilde{Z}_i^\top \in \mathbb{R}^{n \times n}. \tag{9}$$

Here, $(G_i)_{k\ell} = \langle \tilde{Z}_{i,k,:}, \tilde{Z}_{i,\ell,:} \rangle$ is the (unnormalized) inner product between the component vectors $k$ and $\ell$. If different components encode distinct information, these inner products should be small

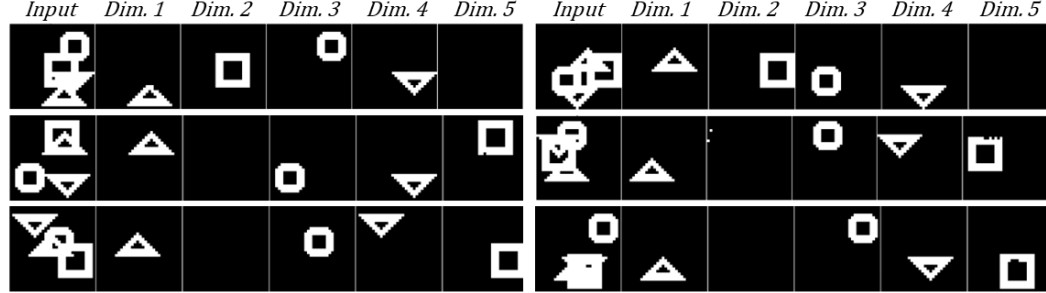

Figure 3: Qualitative OrthoRF results on 4Shapes, after thresholding $\psi_{\text{final}}$. Objects occupy distinct dimensions, and occluded parts are recovered.

(ideally zero) off the diagonal. Then, we penalize the square off-diagonal mass, averaged over samples and unique component pairs:

$$\mathcal{L}_{\text{ortho}} \;=\; \frac{1}{\text{bs}\,n(n-1)} \sum_{i=1}^{\text{bs}} \Big( \text{offdiag}(G_i) \Big)^2 \;=\; \frac{1}{\text{bs}\,n(n-1)} \sum_{i=1}^{\text{bs}} \sum_{k \neq \ell}^{n} (G_i)_{k\ell}^2. \tag{10}$$

By squaring and averaging the $(G_i)_{k\ell}$ terms, we drive cross-component similarities toward zero, thereby decorrelating the embeddings and promoting orthogonality. The factor $n(n-1)$ normalizes by the number of ordered pairs (or twice the number of unique pairs). Finally, the full objective includes the orthogonality term weighted by $\lambda$:

$$\mathcal{L}_{\text{total}} \;=\; \mathcal{L}_{\text{REC}} \;+\; \lambda\, \mathcal{L}_{\text{ortho}}, \qquad \lambda > 0. \tag{11}$$

**Equivariance** The OrthoRF autoencoder exhibits a key property, analogous to SA (Locatello et al., 2020), namely permutation equivariance over orientation components. For a representation $\mathbf{x} \in \mathbb{R}^{\text{bs} \times n \times d}$ (e.g. output of encoder) with $n$ the orientation axis, any permutation $\Pi$ acting on this axis satisfies $f(\Pi \mathbf{x}) = \Pi f(\mathbf{x})$. This property arises from weight sharing across orientation components at every layer, which guarantees identical processing for each component.

**Magnitude gating and occlusion completion** In the final binding step (Eq. 5), the output is $\mathbf{z}_{\text{out}} = m_{\text{out}} \odot \frac{\psi}{||\psi||_2}$, where the magnitude $\mathbf{m}_{\text{out}}$ serves as a visibility gate: visible regions pass, occluded regions are suppressed. The pre-gated content $\psi$ shows occlusion-complete shapes (Fig. 2). A plausible explanation is that $\psi$ is predicted from learned shape priors under the reconstruction objective, yielding completion behind occluders, while $\mathbf{m}_{\text{out}}$ encodes visibility. This selective behavior depends on the softmax over orientation channels (competitive binding), which enables clean gating at the final layer.

## 4 EXPERIMENTS

### 4.1 EXPERIMENTAL SETTINGS

**Datasets** We follow the evaluation protocol of Löwe et al. (2024a; 2022), benchmarking object discovery on datasets with varying numbers of geometric shapes. From RF, we use the 4Shapes dataset (binary variant), containing four object types placed at random locations with possible overlap. We choose the binary form due to RF's reported sensitivity to color variation. We also evaluate on MNIST&Shape (Löwe et al., 2022), pairing one MNIST digit with a randomly positioned geometric shape, and on the Shapes dataset of Miyato et al. (2024), which contains 2–4 objects randomly sampled from four basic shape classes. To assess performance in a more realistic domain, we introduce a synthetic SEM dataset of four stacked material layers, reflecting those used in semiconductor manufacturing, each containing repeated patterns from a single shape class. Layers are laterally shifted and heavily occlude each other. We provide clean and noisy variants to mimic SEM imaging conditions. This dataset emphasizes challenges such as occlusion, layer disentanglement, and robustness to noise. For additional dataset details, see the Appendix.

Table 1: Object discovery on 4Shapes dataset (mean $\pm$ standard deviation over 4 seeds). OrthoRF matches RF under identical output post-processing, but surpasses it on the shape-completion task, measured by $\mathrm{MBO_i^{OV}}$ (OV: overlapping regions), when evaluated on $\psi_{final}$. *MSE, ARI-BG, and MBO$_i$ are taken from Löwe et al. (2024a); RF's ARI-BG and MBO$_i$ were recomputed (see Table 7 Appendix). Best results are highlighted as **first**, second, and third.

| Model | $n$ | $\lambda$ | MSE $\downarrow$ | ARI-BG $\uparrow$ | MBO$_i$ $\uparrow$ | MBO$_i^{OV}$ $\uparrow$ |
|---|---|---|---|---|---|---|
| *AE | - | - | 5.492e-03 $\pm$ 9.393e-04 | - | - | - |
| *CAE | - | - | 3.435e-03 $\pm$ 2.899e-04 | 0.694 $\pm$ 0.041 | 0.628 $\pm$ 0.039 | - |
| *RF$_{z_{final}}^{kmeans}$ | 2 | - | **2.198e-03 $\pm$ 1.110e-04** | **0.667 $\pm$ 0.046** | **0.589 $\pm$ 0.026** | **0.5076 $\pm$ 0.032** |
| | 3 | - | 1.112e-03 $\pm$ 2.698e-04 | **0.937 $\pm$ 0.018** | **0.861 $\pm$ 0.020** | **0.6379 $\pm$ 0.017** |
| | 4 | - | 5.893e-04 $\pm$ 4.350e-05 | 0.944 $\pm$ 0.016 | 0.908 $\pm$ 0.012 | 0.6981 $\pm$ 0.025 |
| | 5 | - | 5.439e-04 $\pm$ 6.984e-05 | 0.975 $\pm$ 0.003 | 0.934 $\pm$ 0.006 | 0.8049 $\pm$ 0.013 |
| | 6 | - | 2.526e-04 $\pm$ 1.416e-05 | 0.991 $\pm$ 0.002 | 0.970 $\pm$ 0.003 | 0.8111 $\pm$ 0.009 |
| | 7 | - | **1.642e-04 $\pm$ 1.810e-05** | 0.992 $\pm$ 0.002 | 0.974 $\pm$ 0.003 | 0.8172 $\pm$ 0.001 |
| | 8 | - | **1.360e-04 $\pm$ 7.644e-06** | 0.987 $\pm$ 0.003 | **0.968 $\pm$ 0.008** | 0.8196 $\pm$ 0.001 |
| | 9 | - | 1.119e-03 $\pm$ 5.715e-04 | 0.9949 $\pm$ 0.002 | **0.989 $\pm$ 0.002** | 0.8170 $\pm$ 0.001 |
| | 20 | - | 1.000e-04 $\pm$ 0.000e-04 | 0.7129 $\pm$ 0.283 | 0.7340 $\pm$ 0.242 | 0.5822 $\pm$ 0.179 |
| OrthoRF$_{z_{final}}^{kmeans}$ | 2 | 0.5 | 3.050e-03 $\pm$ 1.202e-03 | 0.4987 $\pm$ 0.058 | 0.4499 $\pm$ 0.041 | 0.3719 $\pm$ 0.037 |
| | 3 | 0.5 | **1.000e-03 $\pm$ 0.000e-04** | 0.6530 $\pm$ 0.009 | 0.6576 $\pm$ 0.029 | 0.5127 $\pm$ 0.016 |
| | 4 | 0.5 | **4.500e-04 $\pm$ 0.707e-04** | **0.9994 $\pm$ 0.0007** | 0.8915 $\pm$ 0.029 | 0.7205 $\pm$ 0.024 |
| | 5 | 0.8 | **2.330e-04 $\pm$ 0.942e-03** | **0.9995 $\pm$ 0.001** | **0.9887 $\pm$ 0.003** | 0.8204 $\pm$ 0.002 |
| | 6 | 0.5 | **1.963e-04 $\pm$ 0.855e-03** | **0.9941 $\pm$ 0.009** | **0.9856 $\pm$ 0.006** | 0.8151 $\pm$ 0.006 |
| | 7 | 0.1 | 1.660e-04 $\pm$ 0.942e-03 | **0.9947 $\pm$ 0.006** | **0.9856 $\pm$ 0.008** | 0.8161 $\pm$ 0.004 |
| | 8 | 0.09 | 3.714e-04 $\pm$ 0.768e-03 | **0.9924 $\pm$ 0.009** | 0.9527 $\pm$ 0.018 | 0.8022 $\pm$ 0.003 |
| | 9 | 0.08 | **2.133e-04 $\pm$ 0.991e-03** | **0.9955 $\pm$ 0.002** | 0.9849 $\pm$ 0.005 | 0.8183 $\pm$ 0.001 |
| | 20 | 0.1 | **0.003e-05 $\pm$ 0.141e-03** | **0.9974 $\pm$ 0.001** | 0.9764 $\pm$ 0.016 | 0.7426 $\pm$ 0.005 |
| OrthoRF$_{\psi_{final}}^{thresh.}$ | 4 | 0.5 | 4.500e-04 $\pm$ 0.707e-04 | 0.9870 $\pm$ 0.004 | **0.9760 $\pm$ 0.005** | **0.9661 $\pm$ 0.021** |
| | 5 | 0.8 | 2.330e-04 $\pm$ 0.942e-03 | 0.9934 $\pm$ 0.001 | 0.9843 $\pm$ 0.009 | **0.9832 $\pm$ 0.006** |
| | 6 | 0.5 | 1.963e-04 $\pm$ 0.855e-03 | 0.9869 $\pm$ 0.004 | 0.9845 $\pm$ 0.004 | **0.9853 $\pm$ 0.003** |
| | 7 | 0.1 | 1.660e-04 $\pm$ 0.942e-03 | 0.9763 $\pm$ 0.001 | 0.9730 $\pm$ 0.002 | **0.9794 $\pm$ 0.005** |
| | 8 | 0.09 | 3.714e-04 $\pm$ 0.768e-03 | 0.9682 $\pm$ 0.005 | 0.9604 $\pm$ 0.002 | **0.9680 $\pm$ 0.002** |
| | 9 | 0.08 | 2.133e-04 $\pm$ 0.991e-03 | 0.9631 $\pm$ 0.003 | 0.9678 $\pm$ 0.002 | **0.9875 $\pm$ 0.009** |
| | 20 | 0.1 | 0.003e-05 $\pm$ 0.141e-03 | 0.9822 $\pm$ 0.008 | 0.9670 $\pm$ 0.011 | **0.9696 $\pm$ 0.001** |

**Evaluation metrics**   Following standard practices in object discovery, we evaluate performance using the Adjusted Rand Index (ARI) (Hubert & Arabie, 1985; Rand, 1971; Greff et al., 2019) and Mean Best Overlap (MBO) (Pont-Tuset et al., 2016; Seitzer et al., 2022). ARI measures clustering similarity, ranging from 0 (chance) to 1 (perfect match). We compute it using decoder-predicted object masks against ground truth, either excluding background (ARI-BG) or including it (ARI+BG). MBO matches predicted and ground-truth masks by overlap and averages their Intersection-over-Union (IoU) scores.

**Implementation details**   The OrthoRF implementation follows RF (Löwe et al., 2024a) using a convolutional autoencoder. Architectural details appear in Table 9 and Table 10 (Appendix). The models were trained with Adam (Kingma & Ba, 2015) using a batch size of 16 for 100-200k steps across different datasets, with a CosineAnnealingLR scheduler applied for learning-rate decay. Experiments were run in PyTorch (Paszke et al., 2019) on a single NVIDIA Tesla T4 (16 GB). Additional training settings are listed in Table 11 (Appendix).

## 4.2   RESULTS

**Evaluation on the 4Shapes dataset**   Table 1 compares OrthoRF with RF (Löwe et al., 2024a), AE (Löwe et al., 2024a), and CAE (Löwe et al., 2022) on 4Shapes under two protocols. First, visible-only object discovery evaluates model outputs ($z_{final}$ for RF/OrthoRF) against ground-truth masks that exclude overlapping regions (as defined by Löwe et al. (2024a)). Second, we evaluate shape completion using $\mathrm{MBO_i^{OV}}$, which measures full-object recovery—including overlapping (OV) regions—by comparing predictions to instance-level ground-truth masks (labeling scheme in Appendix B.3). In Table 1, we first report RF's $z_{final}$ using its standard post-processing (output normalization, magnitude masking, and $k$-means). We then apply the same post-processing to OrthoRF's $z_{final}$ for a fair comparison. Finally, we evaluate OrthoRF's intermediate map $\psi_{final}$—the

Table 2: Object discovery on the SEM dataset. OrthoRF outperforms RF under severe occlusions and noise, and shows stronger out-of-distribution generalization (noisy testing after clean training, and vice versa). Best results are highlighted as **first**, second, and third.

| Test | Train | Model | $n$ | Noise-free | | | Noisy | | |
|------|-------|-------|-----|------|--------|--------|------|--------|--------|
| | | | | MSE $\downarrow$ | ARI-BG $\uparrow$ | MBO$_i$ $\uparrow$ | MSE $\downarrow$ | ARI-BG $\uparrow$ | MBO$_i$ $\uparrow$ |
| Noise-free | | K-means | - | - | 0.8427 | 0.6546 | - | - | - |
| | | Histogram | - | - | 0.8011 | 0.6582 | - | - | - |
| | | RF$^{\text{kmeans}}_{\mathbf{z}_{\text{final}}}$ | 5 | **0.0001** | 0.9551 | 0.6834 | **0.0057** | 0.6942 | 0.4146 |
| | | OrthoRF$^{\text{thresh.}}_{\psi_{\text{final}}}$ | 5 | 0.0002 | **0.9908** | **0.7171** | 0.0624 | **0.7610** | **0.5644** |
| Noisy | | K-means | - | - | - | - | - | 0.3381 | 0.3442 |
| | | Histogram | - | - | - | - | - | 0.3333 | 0.3612 |
| | | RF$^{\text{kmeans}}_{\mathbf{z}_{\text{final}}}$ | 5 | **0.0042** | 0.8816 | 0.6044 | **0.0007** | 0.7043 | 0.4154 |
| | | OrthoRF$^{\text{thresh.}}_{\psi_{\text{final}}}$ | 5 | 0.0051 | **0.9836** | **0.6705** | **0.0007** | **0.8356** | **0.6268** |

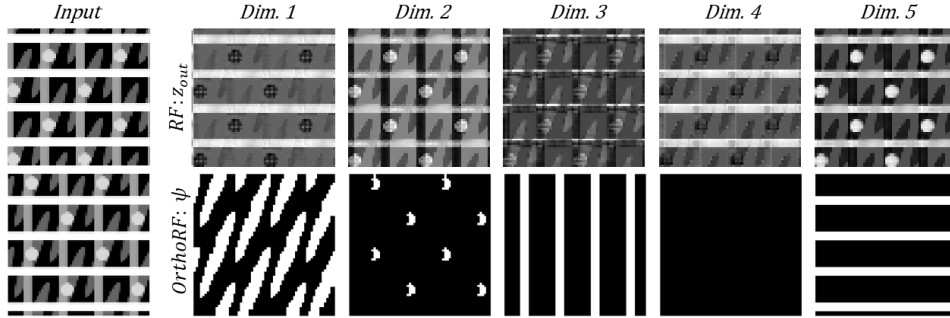

Figure 4: Qualitative results on noise-free SEM test images for $n = 5$. RF's output (top row) is spread across the orientation components, whereas the thresholded OrthoRF's output (bottom row) separates the SEM layers and reveals occluded structures.

gating mechanism's intermediate representation in the output layer—by thresholding it at 0.1 (see Appendix B.4 for details), without any additional post-processing, to obtain binarized masks for metric computation. Note that thresholding is used solely for binarization; the objects in $\psi_{\text{final}}$ are already disentangled, unlike in RF where $k$-means is required.

Table 1 shows that OrthoRF matches RF on visible-only discovery for ARI-BG and MBO$_i$ metrics, but outperforms on shape completion (last column) when evaluated on $\psi_{\text{final}}$. With a global threshold, OrthoRF's $\psi_{\text{final}}$ reaches approximately 0.98 MBO$_i^{\text{OV}}$ at $n = 5$, while the same value for $\mathbf{z}_{\text{out}}$ of both RF and OrthoRF is approximately 0.80. Post-processing largely explains these outcomes: because RF doesn't cleanly separate objects in $\mathbf{z}_{\text{out}}$, it uses $k$-means to recover memberships. However, $k$-means enforces one label per pixel, so overlaps get credit for only a single object. Thresholding $\psi_{\text{final}}$ instead permits multi-label pixels in overlapping regions, improving MBO$_i^{\text{OV}}$. Overall, performance for both models remains stable across $n$ (minor fluctuations). Finally, OrthoRF and RF substantially outperform AE and CAE. Fig. 3 shows qualitative OrthoRF results after thresholding $\psi_{\text{final}}$; objects separate into distinct dimensions, and occluded parts are recovered.

In Table 1, we also compare performance across different orientation dimensionalities $n$. When $n$ is smaller than the number of objects, RF's distributed representations outperform OrthoRF. In contrast, when $n$ is substantially larger (e.g., $n = 20$), OrthoRF maintains high performance, whereas RF shows a clear degradation. For the thresholded $\psi_{\text{final}}$ variant, we omit results when $n$ is smaller than the number of objects, since a one-to-one correspondence between predicted and ground-truth masks is not achievable under these settings. Additional analysis of varying the orthogonality loss weight $\lambda$ is provided in Appendix C.2.

**Evaluation on the SEM datasets** Table 2 reports visible-only object discovery results for OrthoRF, RF, and two non-neural baselines, $k$-means and a histogram-based method, on clean and noisy SEM datasets. For $k$-means, we use scikit-learn's implementation to cluster pixel intensities. The histogram baseline uses mode/peak assignment (nearest-peak clustering); in the intensity his-

Table 3: Object discovery results on MNIST&Shapes (mean $\pm$ standard deviation over 8 seeds). OrthoRF matches RF and CAE on most metrics and achieves the highest ARI-BG, while DBM and SA fail to reliably discover objects. Best results are highlighted as first, second, and third.

| Model | MSE $\downarrow$ | ARI+BG $\uparrow$ | ARI-BG $\uparrow$ |
|---|---|---|---|
| DBM | 1.56e-02 $\pm$ 8.07e-05 | 0.718 $\pm$ 0.002 | 0.175 $\pm$ 0.006 |
| CAE | **3.19e-03 $\pm$ 1.51e-04** | **0.783 $\pm$ 0.007** | 0.971 $\pm$ 0.011 |
| RF | 6.00e-03 $\pm$ 0.00e-04 | 0.699 $\pm$ 0.008 | 0.972 $\pm$ 0.009 |
| OrthoRF | 6.00e-03 $\pm$ 0.00e-04 | 0.667 $\pm$ 0.002 | **0.996 $\pm$ 0.003** |
| SA | 5.44e-03 $\pm$ 1.61e-03 | 0.047 $\pm$ 0.013 | 0.089 $\pm$ 0.028 |

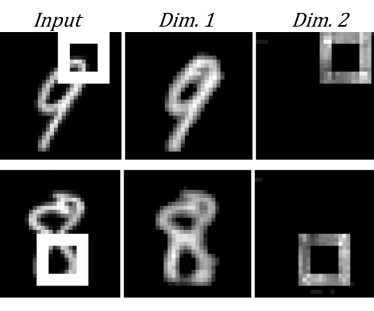

*Input*  *Dim. 1*  *Dim. 2*

Figure 5: Qualitative results of OrthoRF on MNIST&Shapes dataset.

togram, we detect prominent peaks, and assign each pixel to the closest peak. These non-neural baselines were chosen because they are unsupervised, fast, and interpretable. They also show how much segmentation is possible from intensity alone on uniform-color SEM shapes, without heavy architectures, or additional training. Both OrthoRF and RF use $n = 5$ (4 object layers + background) dimensionality in the orientation space.

Table 2 shows that OrthoRF outperforms RF on both the noise-free (ARI-BG: 0.9908 vs 0.9551) and the noisy test set (ARI-BG: 0.8356 vs 0.7043), indicating greater robustness under heavy occlusion and noise. Compared with $k$-means and the histogram baseline, the neural models excel especially in noisy settings, where edge blurring potentially hinders intensity-only clustering. We also evaluated out-of-distribution generalization for OrthoRF. Training on clean data and testing on noisy yields only a minor drop (ARI-BG from 0.9908 to 0.9836), indicating strong noise tolerance. In contrast, training on noisy data and testing on clean degrades more (from 0.8356 to 0.7610), likely because noise-trained models learn smoothed boundaries that underfit sharp, clean edges. Fig. 4 shows qualitative results on noise-free SEM test images. RF (top) distributes content across orientation components, whereas OrthoRF (bottom) separates SEM layers and recovers occluded structures.

**Evaluation on the MNIST&Shape dataset** Since OrthoRF converts distributed representations into discrete ones, we compare it to slot-based methods (SA (Locatello et al., 2020)), synchrony-based approaches (RF, CAE), and the Deep Boltzmann Machine (DBM; (Salakhutdinov & Hinton, 2009)), following the setup of Reichert & Serre (2013); Löwe et al. (2022), on the MNIST&Shape dataset. As shown in Table 3, OrthoRF matches RF's performance while achieving clear object separation (Fig. 5). DBM and SA both fail on MNIST&Shape. SA, in particular, struggles because (1) the MNIST digits exceed its receptive field and (2) it handles grayscale inputs poorly, as reported by Löwe et al. (2022).

Table 4: Object discovery on Shapes (mean $\pm$ standard deviation over 3 seeds). OrthoRF outperforms other models when object count and shape vary randomly. Best results are highlighted as first, second, and third.

| Model | $n$ | ARI-BG $\uparrow$ | MBO$_i$ $\uparrow$ |
|---|---|---|---|
| ItrSA | - | 0.570 $\pm$ 0.055 | 0.348 $\pm$ 0.046 |
| AKOr$N^{\text{attn}}$ | - | 0.713 $\pm$ 0.042 | 0.469 $\pm$ 0.014 |
| RF | 8 | 0.744 $\pm$ 0.079 | 0.780 $\pm$ 0.065 |
| OrthoRF | 8 | **0.833 $\pm$ 0.019** | **0.865 $\pm$ 0.013** |

**Effect of Shape and Count Variability** Table 4 presents results on the Shapes dataset, where both the number of objects (ranging from 2 to 4) and the combination of shape types vary per image. The comparison includes Iterative Self-Attention (ItrSA) and AKOr$N$—two models introduced by Miyato et al. (2024), with the latter being a synchrony-based approach inspired by Kuramoto oscillatory neurons—as well as RF. Among these methods, OrthoRF achieves the best performance. Its discrete representation likely contributes to improved object separation in this challenging, variable setting.

Table 5: Architectural ablations on 4Shapes. Variants differ by the use of softmax with centering (SC) and the orthogonality loss ($\lambda$). Best results are highlighted as first, second, and third.

| Model | SC | $\lambda$ | MSE $\downarrow$ | ARI $\uparrow$ | MBO$_i$ $\uparrow$ |
|---|---|---|---|---|---|
| RF | No | 0 | 0.0005 | 0.9750 | 0.9340 |
| OrthoRF | No | 0.1 | **0.0002** | 0.8530 | 0.8680 |
| | Yes | 0 | 0.0034 | 0.6280 | 0.6880 |
| | Yes | 0.1 | **0.0002** | **0.9995** | **0.9887** |

Table 6: Quantifying similarity in phase space via mean pairwise cosine angles (degrees). On 4Shapes, OrthoRF is near-orthogonal with lower variance; on SEM, RF has a slightly higher mean but larger variance. Best results are highlighted as **first**, and second.

| Data | Model | $\lambda$ | Dim.1 | Dim.2 | Dim.3 | Dim.4 | Dim.5 | Avg. |
|------|-------|-----------|-------|-------|-------|-------|-------|------|
| 4Shapes | RF | - | $46.7 \pm 0.96$ | $80.3 \pm 2.1$ | $64.7 \pm 1.5$ | $67.7 \pm 2.4$ | $87.0 \pm 1.7$ | $69.28 \pm 13.91$ |
| | orthoRF | 0.5 | $88.5 \pm 1.29$ | $88.8 \pm 1.3$ | $89.6 \pm 1.0$ | $89.3 \pm 1.0$ | $78.1 \pm 1.7$ | $86.86 \pm 4.39$ |
| SEM | RF | - | $82.6 \pm 2.6$ | $88.6 \pm 2.6$ | $69.9 \pm 2.1$ | $82.4 \pm 4.7$ | $87.1 \pm 3.3$ | $82.12 \pm 6.57$ |
| | orthoRF | 0.1 | $80.0 \pm 3.7$ | $79.0 \pm 2.2$ | $85.8 \pm 2.4$ | $75.5 \pm 3.1$ | $80.7 \pm 2.2$ | $80.2 \pm 3.32$ |

**Ablation on Architectural Modifications** Table 5 shows an ablation on 4Shapes, assessing the effect of softmax with centering (SC) and the orthogonality loss, controlled by $\lambda$ (where $\lambda = 0$ disables the loss). Neither component alone yields strong performance, but their combination leads to a substantial boost, achieving near-perfect ARI and the highest $\text{MBO}_i$ for $n = 5$.

**Quantitative evaluation of similarity** We quantify how the orthogonality constraint shapes the encoder output $z_{\text{out}}^{enc}$ by averaging pairwise cosine angles across all orientation components (Table 6). On 4Shapes, OrthoRF yields angles near $90°$ with far lower variability than RF (std $4.39$ vs. $13.91$), indicating cleaner phase separation. On SEM, OrthoRF averages $80°$ (softer constraint due to lower $\lambda$), while RF attains a higher mean ($82.12°$) but with greater variance. Across datasets, the background dimension (Dim. 5 in 4Shapes; Dim. 4 in SEM) shows the smallest angles, reflecting weaker distinctiveness but is included in all statistics.

**Quantitative evaluation of separability** To assess the orthogonality constraint in the output $z_{\text{final}}$, we compute inter- and intra-cluster angular metrics following Stanić et al. (2023). The inter-cluster metric measures separation between objects by computing unit-normalized centroids per object and evaluating pairwise centroid angles; angles near $90°$ indicate strong, orthogonal separation. The intra-cluster metric measures compactness by computing the angular deviation of each pixel feature from its object's centroid. Metrics are computed per image and averaged over the 4Shapes dataset. Table 6 shows that OrthoRF achieves consistently lower intra-cluster dispersion than RF, indicating tighter object representations. RF attains a higher mean inter-cluster angle, but with large variability (std. $\approx 23.57°$), suggesting unstable separation. In contrast, OrthoRF produces near-orthogonal inter-cluster angles with substantially lower variance.

To aid interpretation, we visualize per-pixel embeddings from the models' outputs ($z_{\text{final}}$) by projecting them to 2D with PCA and normalizing them on the unit circle, then overlaying class centroids as arrows from the origin (see Fig. 7). OrthoRF forms tighter, better-separated clusters, while RF appears more dispersed—consistent with the quantitative metrics. The background cluster (label 0) is naturally more diffuse.

Figure 6: Separability in phase space on 4Shapes using inter- and intra-cluster metrics. OrthoRF forms tighter clusters, while RF shows higher mean inter-angles but with large variability. Best results are highlighted as **first**, and second.

| Model | Inter-cluster ↑ | | Intra-cluster ↓ |
|-------|-----------------|------|-----------------|
| | Min | Mean | |
| RF | 82.89 | $106.47 \pm 23.58$ | $17.29 \pm 6.38$ |
| OrthoRF | **89.85** | $89.92 \pm 0.067$ | $1.09 \pm 4.38$ |

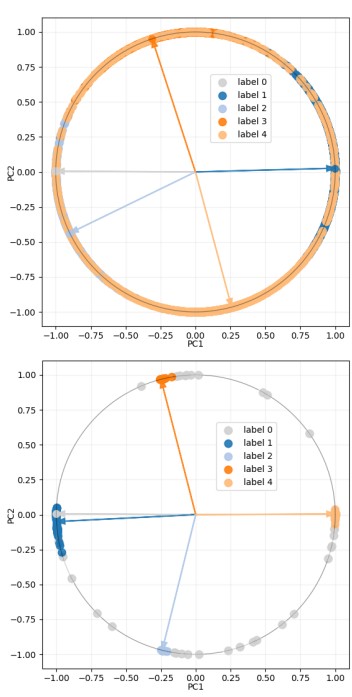

Figure 7: Principal component visualization of $z_{\text{out}}$ on 4Shapes. OrthoRF (bottom) forms tighter clusters, while RF's (top) are more dispersed. Background is always label 0.

## 5 RELATED WORK

**Object-centric learning**    Over the years, a wide range of OCL models (Eslami et al., 2016; Greff et al., 2016; 2017; Jiang et al., 2019; Lin et al., 2020; Prabhudesai et al., 2022; Stelzner et al., 2019) have been proposed. MONet (Burgess et al., 2019), IODINE (Greff et al., 2019), and GENESIS (Engelcke et al., 2019; 2021) propose unsupervised approaches to disentangle scenes into objects, with MONet and IODINE modeling objects independently and GENESIS capturing their interactions. SA (Locatello et al., 2020) extends this line of work by introducing an iterative attention mechanism where slots compete to bind to distinct objects. This design has inspired numerous extensions, including SLATE (Singh et al., 2021) and DINOSAUR (Seitzer et al., 2022), which integrate Transformer-based encoders and decoders (Vaswani et al., 2017) to better handle real-world images. OCL has been extended to both 3D (Chen et al., 2021; Sajjadi et al., 2022; Stelzner et al., 2021) and video data (Lai et al., 2021; Elsayed et al., 2022; Singh et al., 2022). ROOTS (Chen et al., 2021) disentangles objects via 3D-to-2D multi-view projections, while SAVi (Lai et al., 2021), SAVi++ (Elsayed et al., 2022), and STEVE (Singh et al., 2022) extend SA to videos, leveraging temporal dynamics to separate objects from each other and the background. Similarly, PathTracker (Linsley et al., 2021) disentangles objects via trajectory-based grouping, and its extension FeatureTracker (Muzellec et al., 2024) further separates in feature and color space.

**Synchrony-based learning**    Recent OCL works (Löwe et al., 2022; Reichert & Serre, 2013; Löwe et al., 2024a; Stanić et al., 2024) move beyond slot-based models, inspired by neuroscience. While early studies utilized supervised (Mozer et al., 1991) or weakly supervised (Ravishankar Rao & Cecchi, 2010) settings, modern methods emphasize unsupervised discovery. CAE (Löwe et al., 2022) uses complex-valued activations for phase clustering, a framework recently extended via contrastive losses in CtCAE (Stanić et al., 2024) or higher-dimensional rotations in RF (Löwe et al., 2024a). Unlike these models where synchrony emerges dynamically, other works (Miyato et al., 2024; Muzellec et al., 2025) enforce phase synchrony through explicit synchronizers (e.g., Kuramoto oscillators).

**Orthogonality**    Orthogonality has been widely exploited in deep learning. It has been used in network initialization (Saxe et al., 2013) and during training (Achour et al., 2022; Li et al., 2019) to improve stability and generalization. Orthogonality also has been used to create discriminative feature representations (Lezama et al., 2018; Ranasinghe et al., 2021) and disentangle features (Wang et al., 2018). In open-world object detection (Sun et al., 2024), a study utilized multiple levels of orthogonality throughout the training process to mitigate catastrophic interference and facilitate incremental learning of previously unseen objects.

## 6 CONCLUSION

**Summary**    In this paper, we introduce the OrthoRF autoencoder to address a central RF (Löwe et al., 2024a) limitation: distributed object-centric representations break down in overlaps, where features from different objects become uncertain and undermine phase-space clustering. OrthoRF couples competitive binding with an inner-product orthogonality loss to align each object to a distinct phase axis, yielding sharper alignment and removing the clustering step. Across unsupervised object discovery OrthoRF matches or surpasses relevant models and recovers occluded parts. More broadly, orthogonality provides an effective inductive bias that regularizes distributed representations into discrete, directly resolving overlap ambiguity and improving the downstream usability.

**Limitations and future work**    OrthoRF converges more slowly than RF ($\approx$200k vs. $\approx$100k steps), due to its stricter object discovery and separation objective. Rarely, training can get stuck in suboptimal states (incomplete separation or multi-object collapse onto one phase axis), a phenomenon also noted in slot-based setups (Locatello et al., 2020). Increasing $\lambda$ or lowering the learning rate stabilizes training. Future research will explore integrating synchrony-based binding into attention mechanisms, mitigating RF's performance degradation on RGB datasets, and extending evaluations to video and realistic datasets. To offset the computational cost of vector-valued "lifting," we aim to explore more parameter-efficient encodings. Furthermore, inspired by Extreme Image Transforms (EIT) (Crowder & Malik, 2022; Malik et al., 2023), exploring EIT-based augmentations and severe corruptions (e.g., salt-and-pepper noise) remains a promising direction for enhancing robustness.

**Acknowledgements** The presentation of this paper at the conference was financially supported by the Amsterdam ELLIS Unit.

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

APPENDIX

## A    REPRODUCIBILITY STATEMENT

To ensure the reproducibility of our experiments, we provide a detailed overview of the model architectures, hyperparameters, evaluation procedures, datasets, and baselines in Appendix B. While the code is not yet publicly available, we note that our method builds upon RF, allowing others to reproduce our approach by starting from their implementation. The required modifications include adding the orthogonality loss (see Algorithm 1) and implementing changes related to the competitive binding mechanism. All necessary details for the latter are described in the corresponding section of the main paper.

---

**Algorithm 1** Orthogonality Loss Pseudocode

---

**Require:** Batch of rotation encoder outputs $\mathbf{R} \in \mathbb{R}^{B \times N \times D}$, where $B$ is batch size, $N$ is number of slots/rotations, $D$ is feature dimension.
**Ensure:** Scalar loss $\mathcal{L}_{\text{ortho}}$.
    **Center the representations:**
1:  $\boldsymbol{\mu} \leftarrow \frac{1}{N} \sum_{i=1}^{N} \mathbf{R}_{:,i,:}$                                 ▷ Compute mean across the $N$ dimension
2:  $\hat{\mathbf{R}} \leftarrow \mathbf{R} - \boldsymbol{\mu}$                                           ▷ Broadcast subtraction
    **Compute pairwise inner product:**
3:  $\mathbf{C} \leftarrow \text{bmm}(\hat{\mathbf{R}}, \hat{\mathbf{R}}^{\top})$                    ▷ Batch matrix multiplication: $[B, N, N]$
    **Mask and Compute Loss:**
4:  $\mathbf{M} \leftarrow \text{mask}(\mathbf{C}, \neg \mathbf{I})$                        ▷ Select only off-diagonal elements
5:  $\mathcal{L}_{\text{ortho}} \leftarrow \frac{1}{|\mathbf{M}|} \sum_{x \in \mathbf{M}} x^2$           ▷ Mean squared value of off-diagonals
6:  **return** $\mathcal{L}_{\text{ortho}}$

---

## B    IMPLEMENTATION DETAILS

### B.1    ARCHITECTURE AND HYPERPARAMETERS

We implement OrthoRF as a convolutional autoencoder with some fully connected layers, closely following the RF architecture of Löwe et al. (2024a). Below, we specify the exact model configuration and training setup. Tables 9 and 10 provide the architectures used for all datasets and for MNIST&Shape, respectively (using LeakyReLU in place of ReLU). Training hyperparameters are listed in Table 11.

### B.2    DATASET DETAILS

We follow the evaluation setup of Löwe et al. (2024a; 2022), which benchmark object discovery across datasets with varying shape compositions.

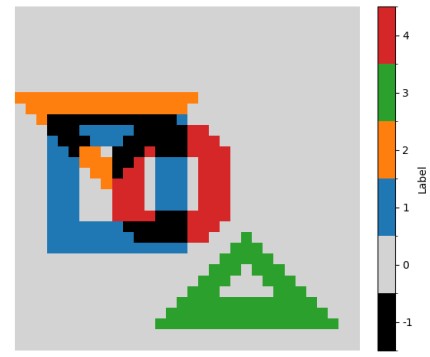

Figure 8: Official label format for the 4Shapes dataset. Occluded regions are labeled with -1 and excluded from evaluation.

**4Shapes**    From the RF paper, we adopt the 4Shapes dataset (see Fig. 3). We use its *binary* variant rather than RGB, motivated by RF's reported sensitivity to color variation: accuracy drops substantially—even with only five colors—when depth masks or pretrained features (e.g., DINO Caron et al. (2021)) are omitted. Each image contains four objects (square, up-pointing triangle, down-pointing triangle, and circle) placed at random locations, with possible overlap and a shared color.

**MNIST&Shape**    Following Löwe et al. (2022), this dataset pairs a single MNIST digit with a randomly positioned geometric shape, increasing both visual diversity and difficulty.

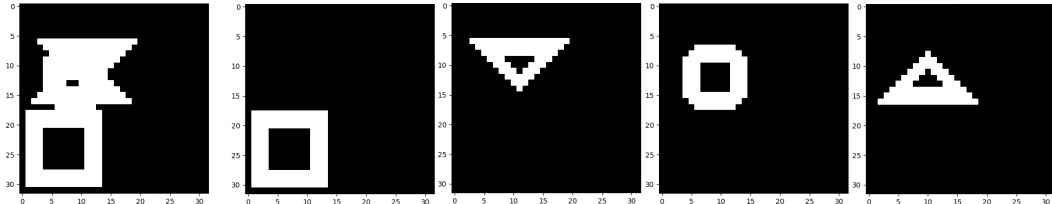

Figure 9: Our labeling scheme used to evaluate occluded regions in the 4Shapes dataset. The leftmost image is the input image and all the rest are binary labels for each of the four shapes.

**Shapes** We include the dataset of Miyato et al. (2024), which generates images with 2–4 objects randomly sampled (with repetition) from triangle, square, circle, and diamond classes.

**SEM Dataset** For a more realistic application domain, we construct a synthetic Scanning Electron Microscope (SEM) dataset of semiconductor materials[1] (see Fig. 4). Each image shows a four-layer vertical stack viewed from above. Each layer contains a single shape class (horizontal lines, circles, vertical lines, or ellipses) repeatedly tiled across the layer. Layers are horizontally displaced relative to one another, and the displacement varies across images. We provide two variants: (i) noise-free and (ii) noisy (Gaussian blur + additive Gaussian noise) to mimic SEM acquisition artifacts.

This dataset is designed to stress-test object-centric models under (i) severe inter-layer occlusion from higher $z$-layers, (ii) relevance to semiconductor metrology, where layer separation is crucial, and (iii) robustness to realistic acquisition noise. While not publicly released, the dataset can be reproduced from the provided description.

### B.3 EVALUATION DETAILS

We use two settings in the our evaluation protocol. (i) Visible-only discovery: we follow Löwe et al. (2024a) and evaluate using their labels, which mark occluded regions with $-1$. This scheme is applied to both binary and RGB data, though we find this setting less fair, especially for boundaries and overlapping regions evaluation. (ii) Shape completion: to assess recovery under occlusion, we create per-shape labels for the 4Shapes dataset, enabling evaluation of both RF and OrthoRF with the $\mathrm{MBO}_{\mathrm{OV}}^{(i)}$ metric. Fig. 8 illustrates the labeling schemes introduced by Löwe et al. (2024a) and Fig. 9 shows our labeling format for the shape completion evaluation.

### B.4 THRESHOLDING

Thresholding is used as a simple post-processing step to binarize the model's output for metric computation. The $\psi$-thresholding procedure is largely insensitive to the choice of threshold. As illustrated in Figs. 10, we first discard background channels (i.e., feature maps with very low variance) and then apply a fixed threshold to the remaining maps. Because competitive binding creates a strong separation between active and inactive regions, a wide range of thresholds produces nearly identical binarizations; 0.1 serves as a convenient default, though other values yield similarly strong results. An adaptive, statistically motivated thresholding scheme is a natural extension for future work.

## C ADDITIONAL EXPERIMENTAL RESULTS

### C.1 EFFECT OF ORTHOGONALITY MECHANISM

Fig. 12 presents three supplementary results from the 4Shapes dataset demonstrating the effect of our modifications in the network. The first row showcases the RF model's output, where object content is distributed across orientations. The second row visualizes the $\psi$ output of the last layer of the OrthoRF model, demonstrating clear object separation and reconstruction of occluded object

---

[1]Public source describing the 3D-layer structure used to generate our synthetic SEM images.

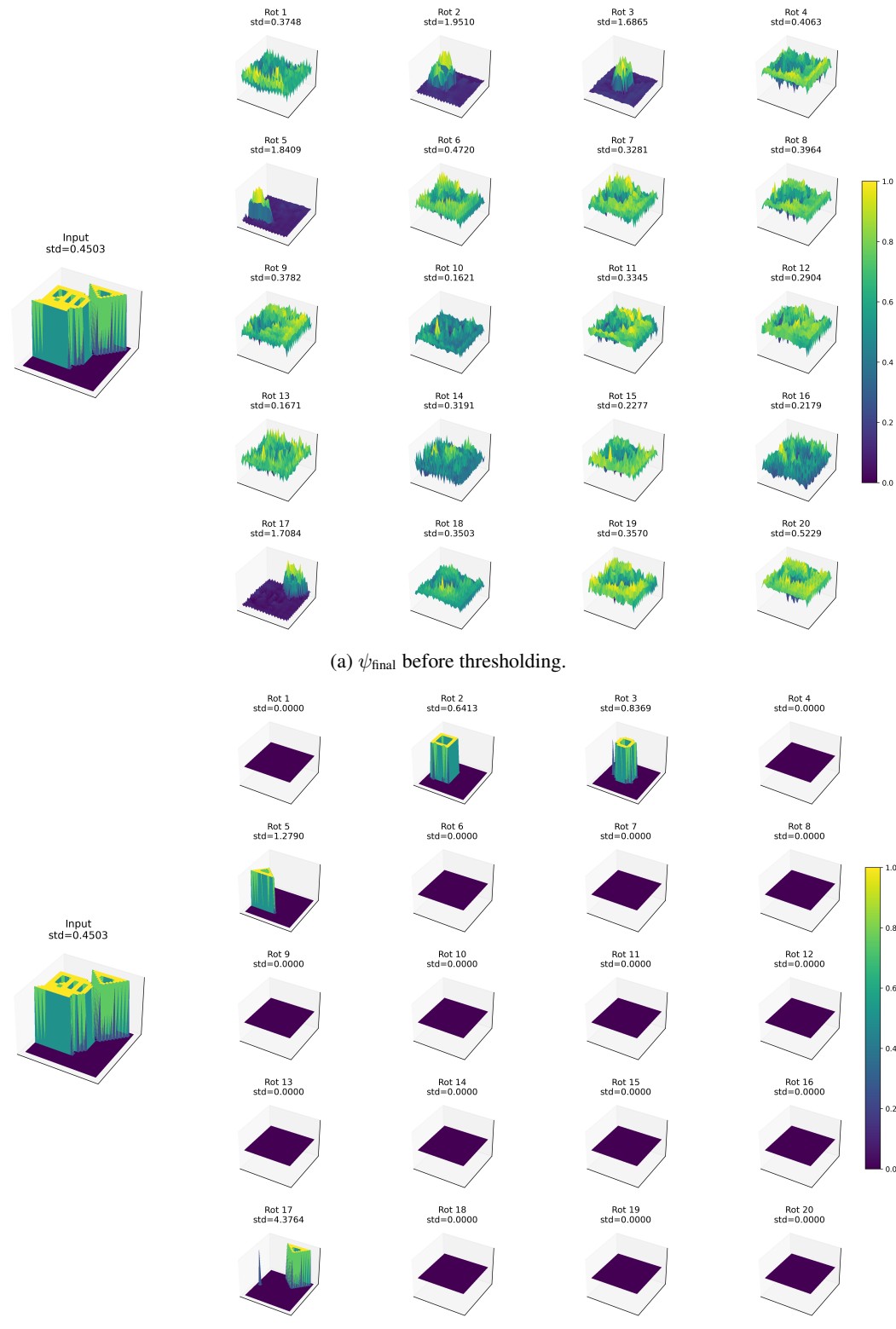

Figure 10: (a) Network output for $n = 20$; object-related dimensions exhibit noticeably higher variance, making them easy to identify. The strong separation between object and background pixels allows a wide range of effective thresholds. (b) Result after applying a fixed threshold to obtain binarized masks.

Table 7: Object discovery on 4Shapes (mean $\pm$ standard deviation over 4 seeds). We reran RF for $n \in \{2, 3, 4, 5, 6, 7, 8\}$ and additionally evaluated $n = 9$. For fair comparison, Table 1 reports the values from Löwe et al. (2024a); our reruns closely match the official results.

| Model | $n$ | ARI-BG $\uparrow$ | MBO$_i$ $\uparrow$ |
|---|---|---|---|
| | 2 | $0.7198 \pm 0.038$ | $0.6526 \pm 0.065$ |
| | 3 | $0.9326 \pm 0.028$ | $0.8615 \pm 0.014$ |
| | 4 | $0.9767 \pm 0.028$ | $0.9382 \pm 0.010$ |
| $^*\text{RF}^{\text{kmeans}}_{z_{\text{final}}}$ | 5 | $0.9931 \pm 0.007$ | $0.9835 \pm 0.007$ |
| | 6 | $0.9950 \pm 0.005$ | $0.9885 \pm 0.006$ |
| | 7 | $0.9980 \pm 0.001$ | $0.9927 \pm 0.001$ |
| | 8 | $0.9979 \pm 0.001$ | $0.9928 \pm 0.001$ |

parts. Finally, the last row depicts the $z_{out}$ output from the OrthoRF's final layer, which reveals object separation but with some holes due to occlusions.

Table 8: Effect of varying orthogonality loss coefficient $\lambda$ and dimensionality $n$ of the orientation space on reconstruction loss, ARI, and MBO.

(a) Reconstruction Loss

| $\lambda$ | 5 | 6 | 7 | 8 | 9 | 20 |
|---|---|---|---|---|---|---|
| 0.05 | 0.0002 | 0.0001 | 0.0001 | 0.0001 | 0.0001 | 0.0002 |
| 0.1 | 0.0002 | 0.0002 | 0.0001 | 0.0001 | 0.0001 | 0.0002 |
| 0.5 | 0.0002 | 0.0001 | 0.0001 | 0.0001 | 0.0002 | 0.0002 |
| 0.8 | 0.0002 | 0.0002 | 0.0001 | 0.0001 | 0.0001 | 0.0002 |
| 0 | 0.0034 | 0.0054 | 0.0043 | 0.0043 | 0.0046 | 0.0001 |

(b) ARI $\uparrow$

| $\lambda$ | 5 | 6 | 7 | 8 | 9 | 20 |
|---|---|---|---|---|---|---|
| 0.05 | 0.8478 | 0.9983 | 0.9996 | 0.9944 | 0.9964 | 0.9971 |
| 0.1 | 0.9997 | 0.9992 | 0.9995 | 0.9994 | 0.9996 | 0.9988 |
| 0.5 | 0.9996 | 0.9912 | 0.9989 | 0.9938 | 0.9806 | 0.9985 |
| 0.8 | 0.9995 | 0.8179 | 0.9992 | 0.9975 | 0.9993 | 0.9433 |
| 0 | 0.6279 | 0.6001 | 0.9676 | 0.7896 | 0.9645 | 0.9975 |

(c) MBO$_i$ $\uparrow$

| $\lambda$ | 5 | 6 | 7 | 8 | 9 | 20 |
|---|---|---|---|---|---|---|
| 0.05 | 0.8872 | 0.8938 | 0.9896 | 0.9832 | 0.9879 | 0.9875 |
| 0.1 | 0.9854 | 0.8758 | 0.9891 | 0.9876 | 0.9887 | 0.9877 |
| 0.5 | 0.9791 | 0.9762 | 0.9853 | 0.9818 | 0.9669 | 0.9859 |
| 0.8 | 0.9843 | 0.8613 | 0.9868 | 0.9880 | 0.9933 | 0.8930 |
| 0 | 0.6881 | 0.6363 | 0.8152 | 0.7071 | 0.8117 | 0.9918 |

## C.2 VARYING $\lambda$

We analyze how performance varies with the orthogonality loss weight $\lambda$ across different numbers of slots $n$. The results in Table 8 and Fig. 11 reveal that for larger $n$, performance remains stable across a wide range of $\lambda$ values. This suggests that higher-dimensional spaces naturally support object separation. However, for smaller $n$ (e.g., $n = 5, 6$), low $\lambda$ values lead to significantly worse ARI and MBO, indicating that orthogonality constraints become essential when representational capacity is limited.

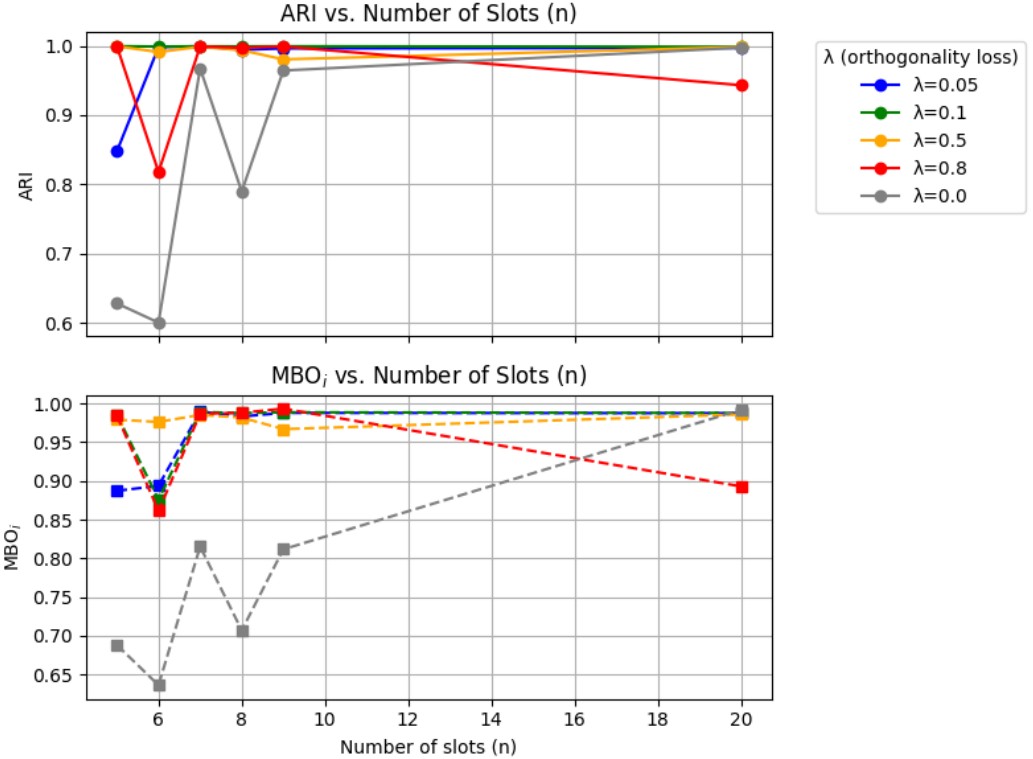

Figure 11: Effect of varying $\lambda$ and number of slots $n$ on ARI and MBO. With larger $n$, performance remains high across $\lambda$ values, while for smaller $n$, sufficient orthogonality regularization ($\lambda > 0$) is crucial for object separation.

### C.3 QUANTITATIVE RESULTS FOR THE 4SHAPES DATASET

In Table 1, we report RF results on 4Shapes as published by Löwe et al. (2024a). For a fair comparison in our main results, we also replicated their experiments; the outcomes are shown in Table 7. Our rerun closely matches the official numbers.

### C.4 QUALITATIVE RESULTS FOR THE 4SHAPES DATASET

In Table 1, we present results of OrthoRF model in comparison with other models for the 4Shapes dataset. Specifically, we vary the orientation-space dimensionality and find that performance remains largely stable. In this section, we show qualitative results for orientation-space dimensionalities $n \in \{6, 7, 8, 9\}$ in Figs. 13–16. The visualizations demonstrate that only the necessary number of dimensions are used while the rest stay empty.

### C.5 QUALITATIVE RESULTS FOR THE SEM DATASET

This section presents qualitative results for the noise-free and noisy SEM datasets. Fig. 17 compares the $z_{out}$ output of the OrthoRF and RF models trained and tested on noise-free SEM images. The leftmost image in each row represents the input image. As shown, the RF model distributes layer content across dimensions, while the OrthoRF model effectively separates SEM layers into distinct dimensions. Fig. 18 displays the OrthoRF output alongside its thresholded version, used for binarization in metric calculations. Fig. 19 shows qualitative results for OrthoRF and RF trained and tested on the noisy SEM dataset. As in the noise-free case, RF distributes content across orientations, whereas OrthoRF cleanly separates layers into distinct dimensions.

### C.6 QUALITATIVE RESULTS FOR THE SHAPES DATASET

In this section, we present qualitative results for orientation-space dimensionality $n = 8$ on the Shapes dataset (see Fig. 21). The visualizations show that OrthoRF activates only the necessary dimensions, while the remaining ones remain unused, reflecting sparse object representations.

### C.7 EFFECT OF MAGNITUDE

Fig. 20 presents the $\psi$ and $z_{out}$ outputs of the OrthoRF model. Notably, the $z_{out}$ output exhibits gaps in the shapes of deeper layers, such as the ellipsoid layer. However, the $\psi$ output (top row) provides information about these missing parts, enabling more complete layer reconstruction.

## D USE OF LARGE LANGUAGE MODELS

We used large language models solely to polish the writing. All scientific claims and analyses were conceived and validated by the authors.

Table 9: OrthoRF architecture with input dimensions $h, w, c$ and feature dimension $d$. All fractions are rounded up. We followed the same architecture as in RF Löwe et al. (2024a).

|  | Layer | Feature Dimension | Kernel | Stride Input / Output | Padding |
|---|---|---|---|---|---|
|  | Input | $h \times w \times c$ | - | - | - |
|  | Conv | $h/2 \times w/2 \times d$ | 3 | 2 | 1 / 0 |
|  | Conv | $h/2 \times w/2 \times d$ | 3 | 1 | 1 / 0 |
|  | Conv | $h/4 \times w/4 \times 2d$ | 3 | 2 | 1 / 0 |
| **Encoder** | Conv | $h/4 \times w/4 \times 2d$ | 3 | 1 | 1 / 0 |
|  | Conv | $h/8 \times w/8 \times 2d$ | 3 | 2 | 1 / 0 |
|  | Reshape | $1 \times 1 \times (h/8 * w/8 * 2d)$ | - | - | - |
|  | Linear | $1 \times 1 \times 2d$ | - | - | - |
|  | Linear | $1 \times 1 \times (h/8 * w/8 * 2d)$ | - | - | - |
|  | Reshape | $h/8 \times w/8 \times 2d$ | - | - | - |
|  | TransConv | $h/4 \times w/4 \times 2d$ | 3 | 2 | 1 / 1 |
| **Decoder** | Conv | $h/4 \times w/4 \times 2d$ | 3 | 1 | 1 / 0 |
|  | TransConv | $h/2 \times w/2 \times d$ | 3 | 2 | 1 / 1 |
|  | Conv | $h/2 \times w/2 \times d$ | 3 | 1 | 1 / 0 |
|  | TransConv | $h \times w \times d$ | 3 | 2 | 1 / 1 |

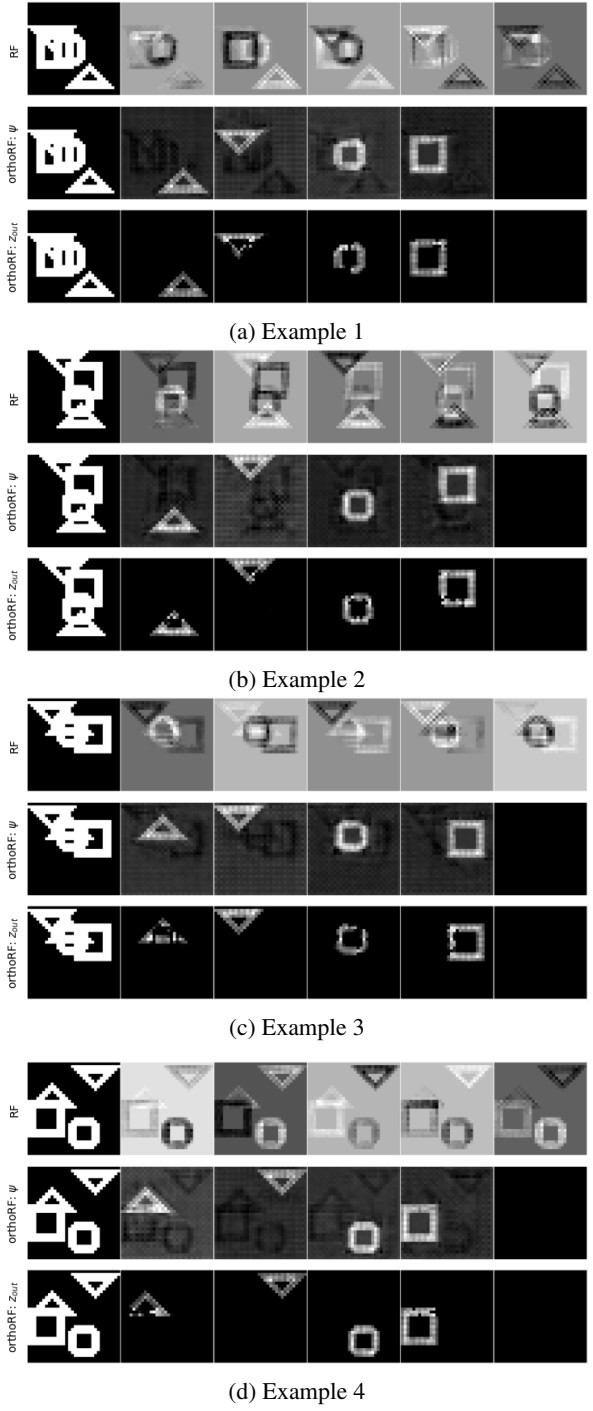

(a) Example 1

(b) Example 2

(c) Example 3

(d) Example 4

Figure 12: Comparison of OrthoRF and RF performance for $n = 5$. Row 1: RF (Löwe et al., 2024a) displays distributed object content. Row 2: OrthoRF's $\psi$ displays distinct objects, including occluded regions. Row 3: OrthoRF's $z_{out}$ shows distinct objects; holes indicate occluded regions.

Table 10: OrthoRF architecture with input dimensions $(h, w, c)$ and feature dimension $d$, presented for the MNIST&Shape dataset. All fractional values are rounded upward.

|  | Layer | Feature Dimension | Kernel | Stride Input / Output | Padding |
|---|---|---|---|---|---|
|  | Input | $h \times w \times c$ | - | - | - |
| **Encoder** | Conv | $h/2 \times w/2 \times d$ | 5 | 2 | 1 / 0 |
|  | Conv | $h/2 \times w/2 \times d$ | 5 | 1 | 1 / 0 |
|  | Conv | $h/4 \times w/4 \times 2d$ | 5 | 2 | 1 / 0 |
|  | Conv | $h/4 \times w/4 \times 2d$ | 5 | 1 | 1 / 0 |
|  | Conv | $h/4 \times w/4 \times 2d$ | 5 | 1 | 1 / 0 |
|  | Reshape | $1 \times 1 \times (h/4 * w/4 * 2d)$ | - | - | - |
|  | Linear | $1 \times 1 \times 2d$ | - | - | - |
| **Decoder** | Linear | $1 \times 1 \times (h/4 * w/4 * 2d)$ | - | - | - |
|  | Reshape | $h/4 \times w/4 \times 2d$ | - | - | - |
|  | TransConv | $h/4 \times w/4 \times 2d$ | 5 | 1 | 1 / 0 |
|  | Conv | $h/4 \times w/4 \times 2d$ | 5 | 1 | 1 / 0 |
|  | TransConv | $h/2 \times w/2 \times d$ | 5 | 2 | 1 / 1 |
|  | Conv | $h/2 \times w/2 \times d$ | 5 | 1 | 1 / 0 |
|  | TransConv | $h \times w \times d$ | 5 | 2 | 1 / 1 |

Table 11: Hyperparameters of the OrthoRF for the five different datasets.

| Dataset | SEM | noisy SEM | 4Shapes | MNIST&Shape | Shapes |
|---|---|---|---|---|---|
| Training Steps | 200k | 200k | 200k | 100k | 100k |
| Batch Size | 16 | 16 | 16 | 16 | 16 |
| LR Warmup Steps | - | - | 1000 | 1000 | 1000 |
| Peak LR | 0.001 | 0.001 | 0.001 | 0.001 | 0.001 |
| Gradient Norm Clipping | 0.1 | 0.1 | 0.1 | 0.1 | 0.1 |
| Feature dim $d$ | 64 | 64 | 64 | 128 | 128 |
| Rotating dim $n$ | 5 | 5 | 5-9 | 2 | 8 |
| Image Size | 256 | 256 | 32 | 32 | 40 |
| Crop Size | 200 | 200 | 32 | 32 | 40 |
| Cropping Strategy | CenterCrop | CenterCrop | - | - | - |
| Rescale Size | 64 | 64 | 32 | 32 | 40 |
| Augmentations | - | - | - | - | - |
| Objects/Image | 4 | 4 | 4 | 2 | 2-4 |
| No. Train Images | 5k | 10k | 50k | 50k | 40k |
| No. Test Images | 6400 | 6400 | 10000 | 10000 | 1000 |

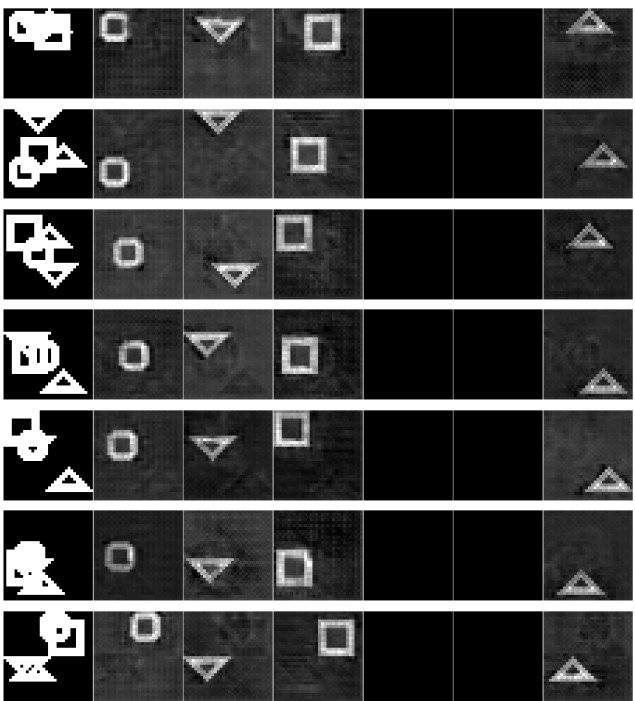

Figure 13: Qualitative results of OrthoRF's $\psi_{final}$ for $n = 6$.

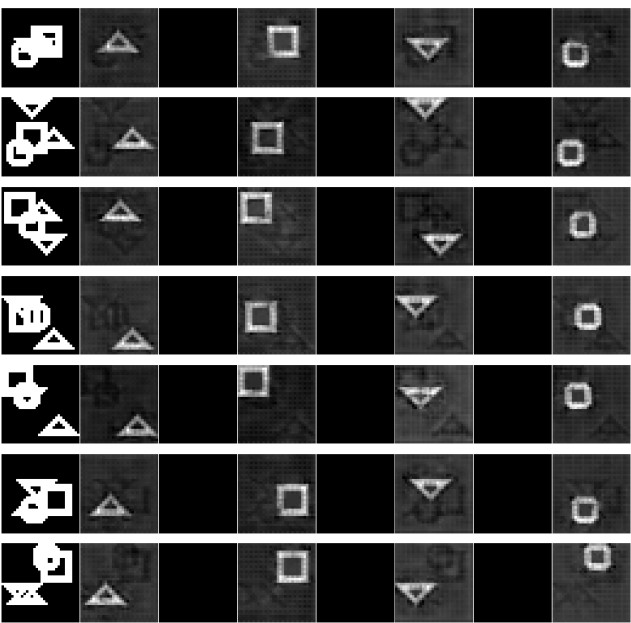

Figure 14: Qualitative results of OrthoRF's $\psi_{final}$ for $n = 7$.

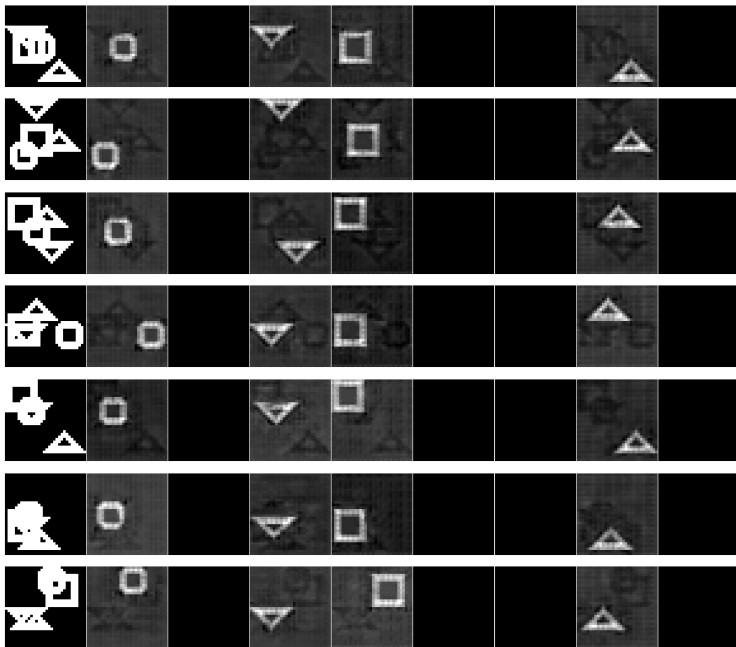

Figure 15: Qualitative results of OrthoRF's $\psi_{final}$ for $n = 8$.

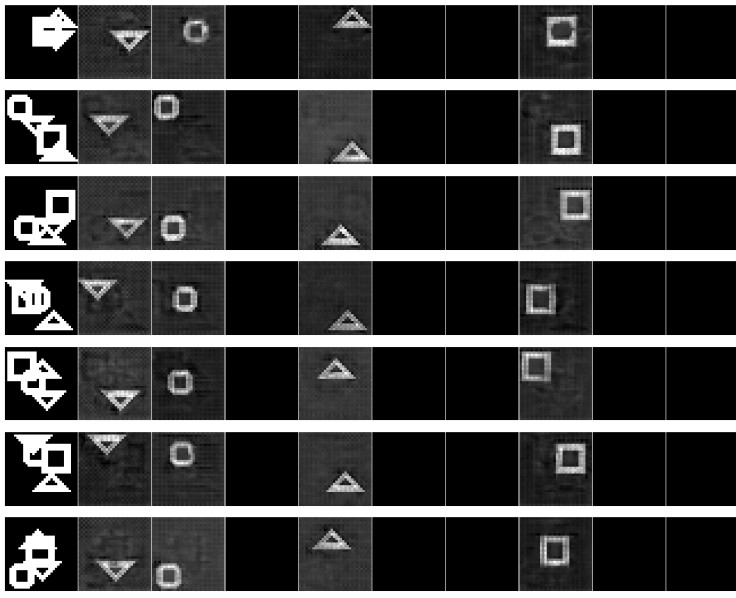

Figure 16: Qualitative results of OrthoRF's $\psi_{final}$ for $n = 9$.

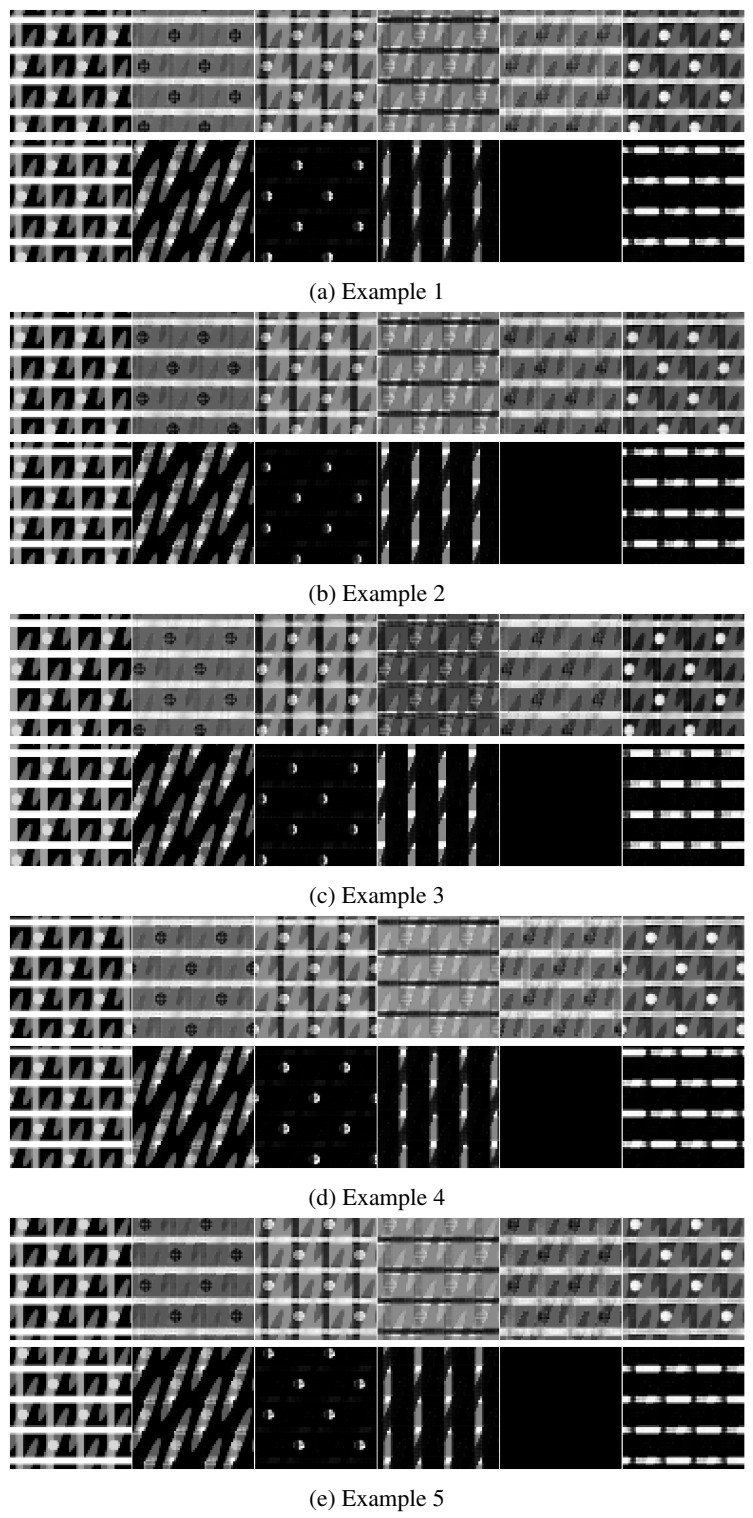

(a) Example 1

(b) Example 2

(c) Example 3

(d) Example 4

(e) Example 5

Figure 17: Qualitative comparison between RF (top row) and OrthoRF (bottom row) for $n = 5$ when trained and tested in the noise-free SEM dataset. We used $z_{final}$ for a fair comparison. The leftmost image in every row is the input image.

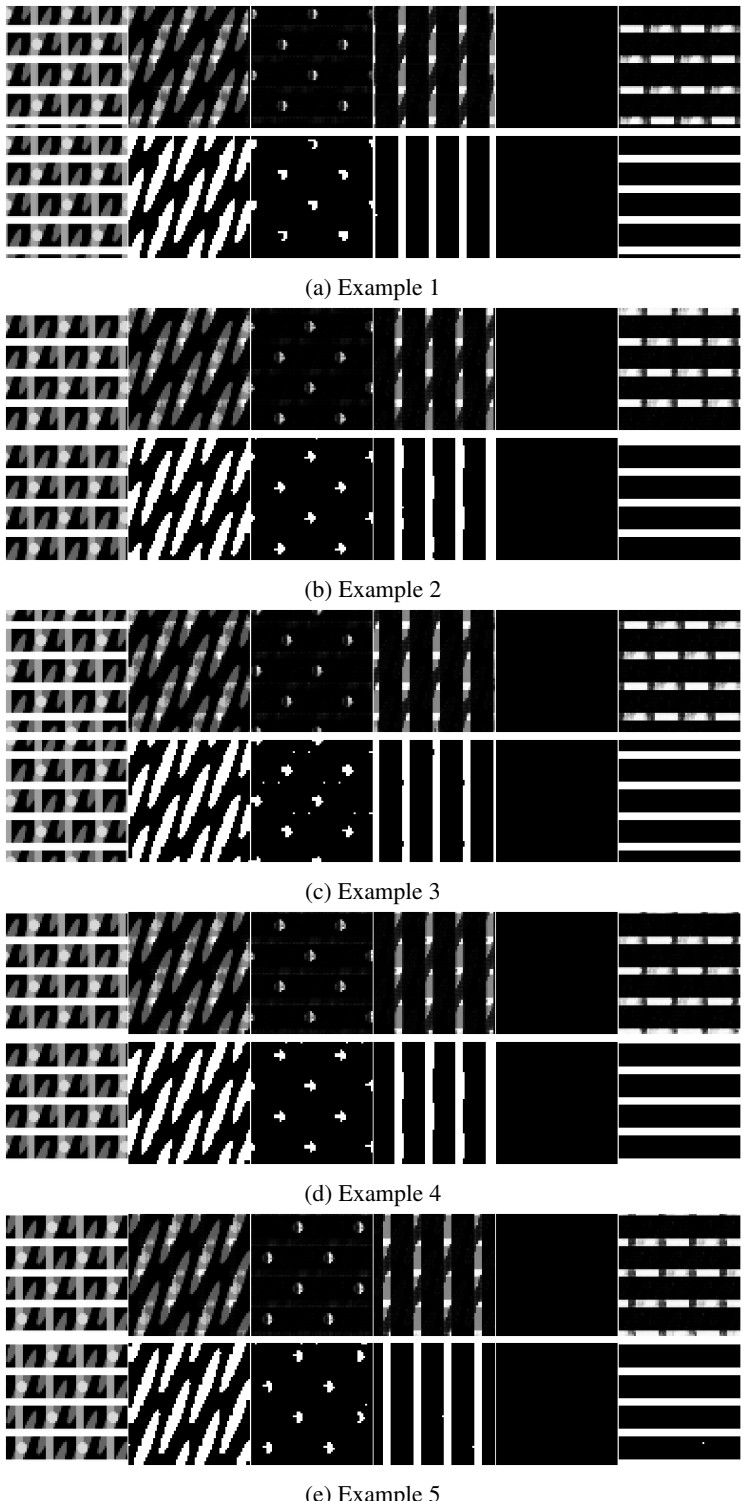

(a) Example 1

(b) Example 2

(c) Example 3

(d) Example 4

(e) Example 5

Figure 18: Qualitative results of OrthoRF for $n = 5$. The top row displays the model's output ($z_{final}$), while the bottom row shows the thresholded rotations that are used for the metrics calculation. The leftmost image in every row is the input image.

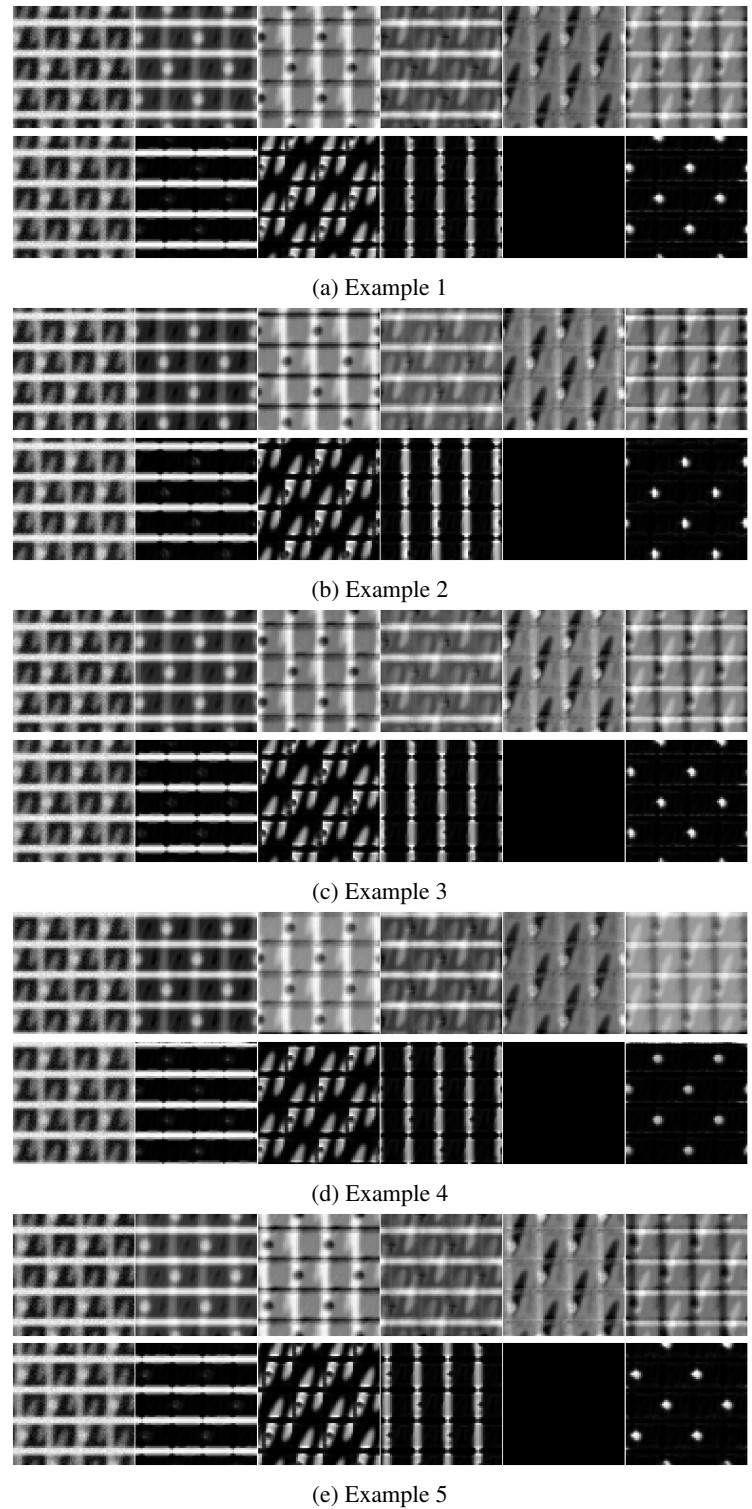

(a) Example 1

(b) Example 2

(c) Example 3

(d) Example 4

(e) Example 5

Figure 19: Qualitative comparison between of RF (top row) and OrthoRF (bottom row) for $n = 5$ when trained and tested in the noisy dataset. We used $z_{final}$ for a fair comparison. The leftmost image in every row is the input image.

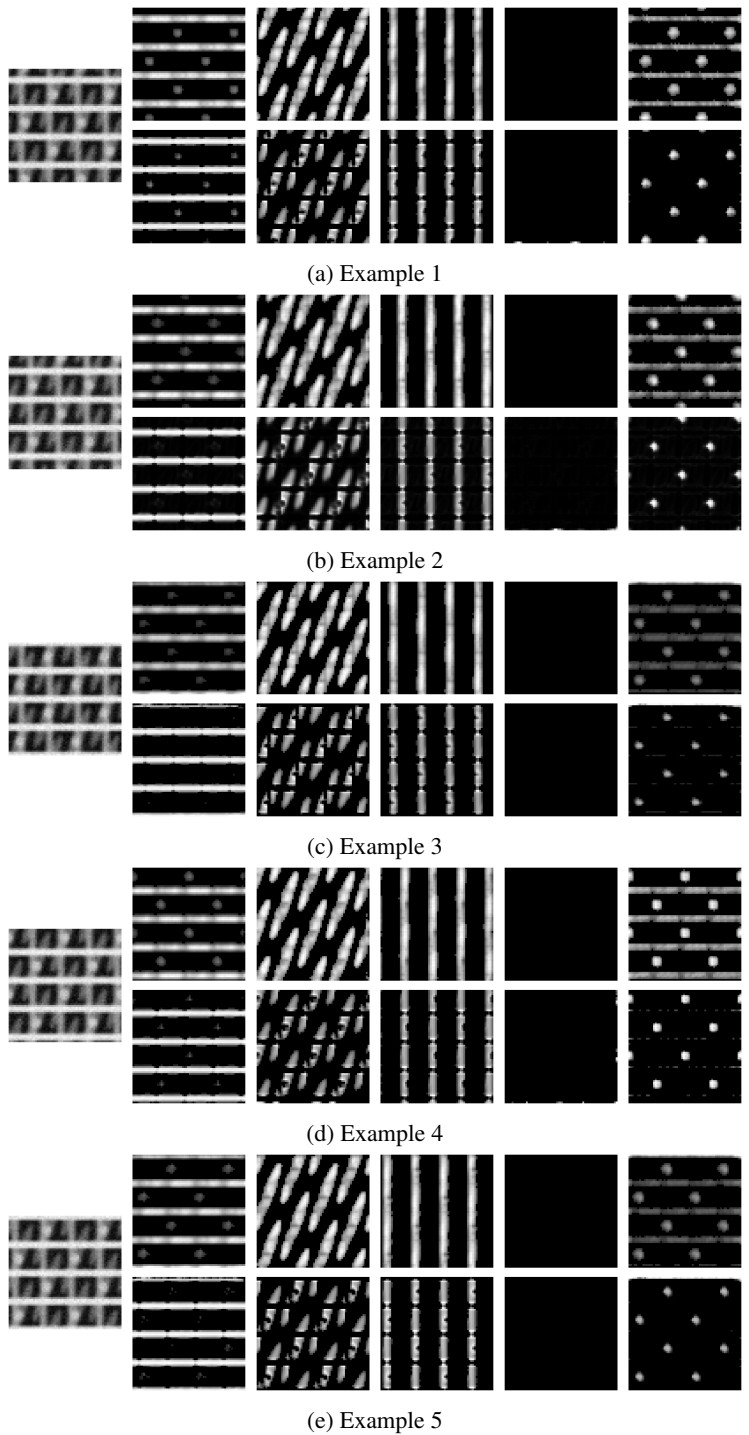

(a) Example 1

(b) Example 2

(c) Example 3

(d) Example 4

(e) Example 5

Figure 20: Qualitative results of OrthoRF for $n = 5$. The top row displays the $\psi_{final}$ and the bottom row $z_{final}$. The $\psi_{final}$ in this case serves to fill the gaps for a layer such as the ellipsoids layer. The leftmost image in every row is the input image.

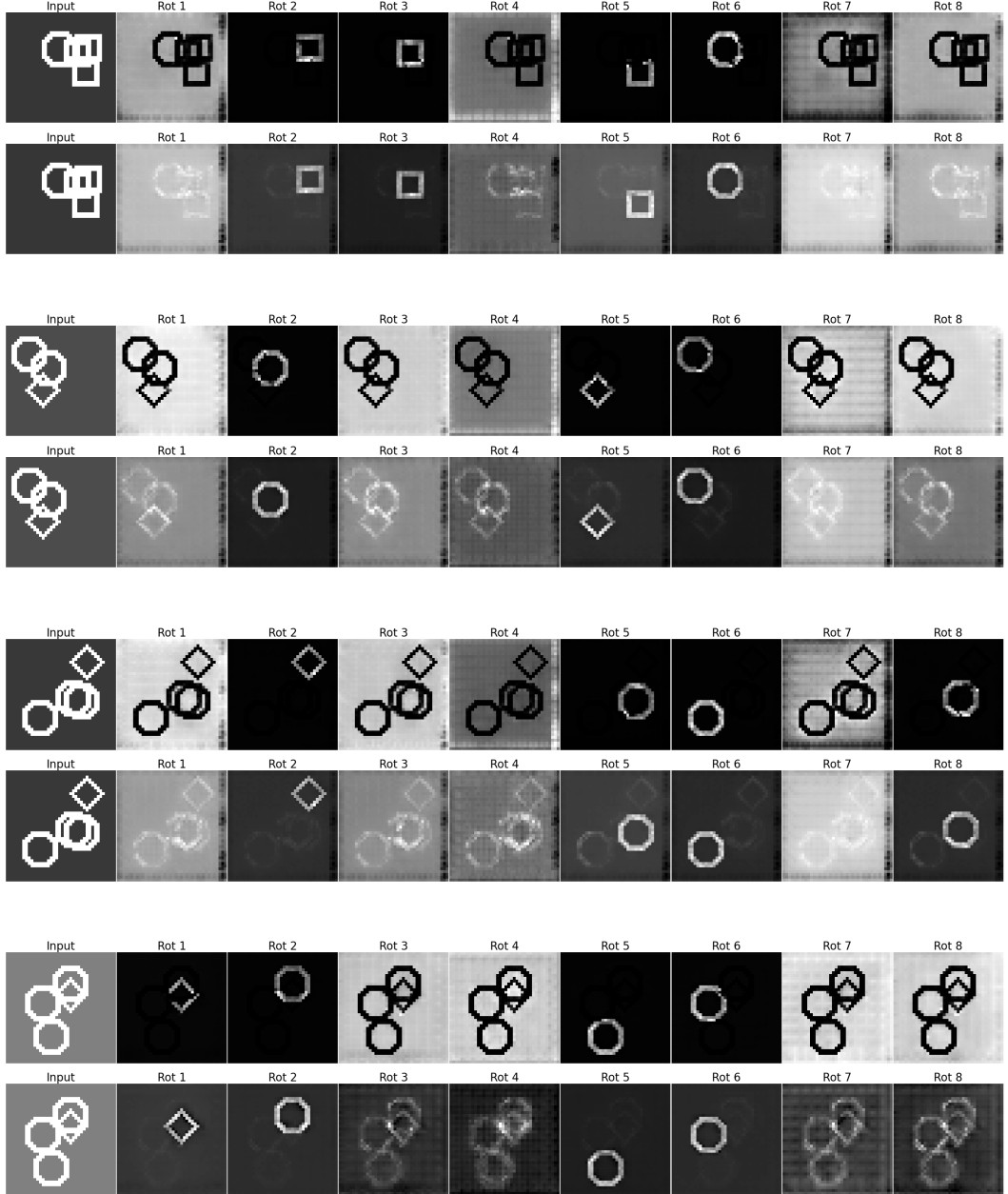

Figure 21: Visualizations of OrthoRF's $z_{\text{final}}$ (top of each pair) and $\psi_{\text{final}}$ (bottom of each pair) for samples from the Shapes dataset.

