# OpenReview forum: "OrthoRF: Exploring Orthogonality in Object-Centric Representations"
_ICLR.cc/2026/Conference — ICLR 2026 Poster_

### Official Review · Reviewer_WoQj · 2025-10-29

**Soundness:** 2
**Presentation:** 3
**Contribution:** 2
**Rating:** 2
**Confidence:** 4

**Summary:**

The authors introduce OrthoRF, an extension of RotatingFeatures improving the state-of-the-art in unsupervised object discovery using synchrony-based representations. This improvement consists in enforcing orthogonality between the different latent spaces of the model, yielding internal object separation akin to Slot-based models. The authors evaluate the performance of their model on 2 synthetic datasets: 4Shapes and SEM and show that OrthoRF tends to outperform RF.

**Strengths:**

**Timely and important problem.** Object-centric learning is a central goal in vision; advancing synchrony-based models is particularly valuable because they promise binding without supervision or heavy architectural machinery.

**Clarity and presentation.** The paper is overall clear and reads well; the motivation, method, and experimental setup are easy to follow.

**Simple, intuitive, and promising idea.** Orthogonality serves as a practical geometric surrogate that encourages non-interfering latent spaces. It’s easy to implement,  and serves as a strong regularizer that constrains networks toward clean, object-separated representations.

**Good quantitative evaluation of learned representation.** The similarity and separability analyses (Section 4.2) give a concrete, interpretable view of internal representation geometry. These measurements constitute useful standardized evaluation practices for synchrony in future work, beyond task metrics alone.

**Weaknesses:**

*Major*

**Loss of flexible dynamic binding.** A core weakness of slot-based methods is their fixed capacity: a preset slot budget limits generalization when scenes contain fewer or more objects than during training. Synchrony-based models explicitly aim to avoid this by distributing object clusters in a continuous phase space, enabling dynamic, per-image allocation. By enforcing orthogonality across latent components, OrthoRF effectively funnels objects back into component-specific subspaces, re-introducing slot-like rigidity and sacrificing one of synchrony’s key advantages.

Is it possible to add empirical comparisons with slot-based models showing that OrthoRF still performs better? For example: the authors could train OrthoRF on 4Shapes and evaluate with 1–8 (or more?) objects per image to demonstrate that OrthoRF can still outperform slot-based solutions when object count varies. Additionally, give the striking similarity between OrthoRF and slot-models, could the authors include the latter as baseline in addition to RF?

**Decoder comparability on 4Shapes.** OrthoRF does not clearly outperform RF on 4Shapes (especially considering RF’s performance in Table 5). The gains appear mainly when thresholding psi_final for OrthoRF while using k-means on z for RF, and performance is better only when including overlapping regions in the evaluation. This mixes architectural effects with decoder choices.

Can RF be evaluated in the same way—thresholding psi—so that both models use comparable decoding strategies, on both datasets? Or could the authors propose ways to cleanly separate improvements due to OrthoRF’s architecture from those due to decoding?

**Benchmark scope and claim accuracy.** The Stanić et al., 2023 evaluation benchmark cited does not include 4Shapes (contrary to the statement at L268); their datasets are high-resolution, colored images (Tetrominoes, dSprites, and CLEVR). Later synchrony work (e.g., Gopalakrishnan, 2024) also reports strong phase separation on those more complex datasets. RF, even though includes some tests on 4Shapes, is also evaluated on colored naturalistic scenes (Pascal, FoodSeg). While simplified grayscale setups can be valuable diagnostically, most work on unsupervised object discovery with synchrony-based models since 2023 evaluates on RGB images, so returning to grayscale toy shapes can feel not fully aligned with current practice; framing this choice explicitly and adding at least one RGB benchmark would help align with field norms.

Could the authors (i) clarify/correct the claims, (ii) either evaluate OrthoRF on the standard color/RGB benchmarks or demonstrate that prior synchrony models specifically fail under the proposed overlap/noisy regimes, (iii) discuss scalability to known failure modes, especially same-color object collisions in RGB (a classic challenge for synchrony). Does orthogonality alleviate phase ambiguity when two objects share hue/texture, or does it merely enforce cleaner axes without resolving color-based binding failures?

*Minor*

**Notation consistency.** The authors define z_out in Eq. 5, then switch to z_final in the subsequent sentence and Eq. 6, and the two are used interchangeably elsewhere. Please clarify whether they are identical; if yes, standardize on one term throughout (update figures/captions accordingly). If not, define z_final precisely and maintain the distinction.

**Typo.** Table 2: “thesh” → “thresh”.

**Capacity fairness.** In Table 1, AE/CAE may not be directly comparable to RF/OrthoRF if they have fewer parameters. Please report parameter counts and, ideally, include capacity-matched baselines or parameter-controlled ablations (and note any compute/step differences) to ensure that performance differences are not confounded by model size.

**Questions:**

**Noise Robustness.** Gaussian blur and additive Gaussian noise may preserve the structure of the image. Have the authors tested more disruptive corruptions (e.g. salt-and-pepper)? Also, do the results still hold under various SNR?

**Figure 2 (n=5).** The figure indicates n=5, seemingly tying the number of channels to the number of slots. Why is this augmentation necessary?
Also, in Figure 7, the panel shows one full scene plus four single-object images. Are these single-object images used only for evaluation (e.g., to compute metrics) or also during training as part of the n channels? If used in training, how are they generated without label leakage, and what supervision signal (if any) do they introduce?

---

> ### Author Response · Authors · 2025-11-25
>
> Thank you for your detailed review and the constructive feedback you provided. We address your points below. We apologize for the short delay, as we conducted additional experiments to ensure a comprehensive response.
>
> 1. **By enforcing orthogonality, OrthoRF may lose synchrony’s ability to dynamically allocate components when the number of objects varies.**
> We appreciate this concern and provide empirical evidence showing that OrthoRF retains flexibility across varying object counts. First, in the undercomplete regime (n < 4), OrthoRF behaves predictably and remains competitive with RF. As shown below, RF is indeed stronger for (n=3,2), but OrthoRF surpasses and reaches good object separation at (n=4):
>
> **Table — GRAY-4Shapes (OrthoRF vs RF)**
>
> | n | OrthoRF ARI | RF ARI | OrthoRF MBO | RF MBO |
> | - | ----------- | ------ | ----------- | ------ |
> | 2 | 0.54        | 0.67   | 0.48        | 0.59   |
> | 3 | 0.65        | 0.94   | 0.68        | 0.86   |
> | 4 | 0.99        | 0.94   | 0.87        | 0.91   |
>
> More importantly, OrthoRF remains stable even when n is far larger than the true number of objects. For example, with (n=20) (5× the actual object count), OrthoRF achieves **ARI = 0.99, MBOi = 0.988** (see Table A in the response to Reviewer 1 with ID wHKK), while RF drops significantly (**ARI = 0.757, MBOi = 0.774**). In these overcomplete regimes, OrthoRF cleanly suppresses unused channels and assigns objects to distinct components without requiring a fixed “slot budget,” whereas RF exhibits phase mixing as the dimensionality grows.
>
> 2. **Comparison with Slot Attention / variable object count.**
> Among non-RF baselines, **Slot Attention** is the most comparable for static OCL. We therefore include results on MNIST&Shape (CAE’s dataset), showing that synchrony-based methods remain competitive and that OrthoRF additionally recovers occluded object parts (visuals will be added to the supplementary material).
>
> | Model          | ARI+BG     | ARI–BG     |
> | -------------- | ---------- | ---------- |
> | CAE            | 0.783      | 0.971      |
> | RF             | 0.6999     | 0.9720     |
> | DBM            | 0.718      | 0.175      |
> | Slot Attention | 0.047      | 0.089      |
> | **OrthoRF**    | **0.6385** | **0.9927** |
>
> Regarding evaluation with varying numbers of objects (1–8 or more), we agree that this is valuable. We are currently running such experiments and are striving to communicate the results soon.
>
> 3. **Decoder Comparability on 4Shapes.**
>
> The key advantage of OrthoRF is that it produces structured, component-wise object representations, which makes post-hoc k-means unnecessary. While RF requires k-means to separate objects and must be provided the number of clusters, OrthoRF yields distinct object maps directly, allowing simple thresholding only for binarization when computing metrics. To clarify comparability: in Table 1 we already evaluate *OrthoRF’s final z output* using the **same k-means decoding** as RF, and the performance is comparable. The improvements arise when using OrthoRF’s ψ maps, which provide clean object-specific channels and enable occlusion recovery — something RF cannot achieve even under identical decoding. Thresholding does not “find” objects; it only converts the already separated ψ channels into binary masks. Moreover, OrthoRF offers a simple unsupervised way to identify unused channels: they have near-zero variance, with a clear gap from the object-bearing ones. In summary, we do provide decoding-matched comparisons (k-means on z for both models), and the additional benefits of ψ-thresholding reflect architectural gains in OrthoRF rather than differences in decoding (see also the first table in the response to reviewer wHKK).

---

> > ### Author Response · Authors · 2025-11-25
> >
> > 4. **Benchmark scope and claim accuracy.**
> > Thank you for raising this. We clarify the points as follows:
> >
> > (i) Correcting the claim.
> > Our intention in L268 was to refer to the evaluation protocol of Stanić et al., 2023 (i.e., ARI-BG, MBO, etc.), not to imply that 4Shapes is part of their benchmark. We will revise the text to avoid this ambiguity and make clear that our datasets differ from those used in their RGB benchmarks.
> >
> > (ii) RGB benchmarks and synchrony-model performance.
> > We agree that most recent work evaluates synchrony-based models on RGB datasets such as Tetrominoes, dSprites-RGB, and CLEVR. However, existing synchrony models, including RF, still experience well-known difficulties in these settings, especially when objects share similar colors or textures. For example, Gopalakrishnan et al. (2023) (SynCx) report ARI scores of ~0.59 on CLEVR, and RF achieves around 0.65 ARI. These limitations motivate our focus: understanding and improving the underlying grouping mechanisms rather than pushing performance on datasets where color collisions remain a dominant failure mode. Our results highlight an orthogonal and under-explored aspect of synchrony-based methods, namely, robust grouping under heavy overlap and occlusion. We show that OrthoRF improves representational structure and eliminates the need for k-means post-processing. This is also practically relevant because in RGB benchmarks like dSprites-RGB or CLEVR the number of objects varies across images; using a fixed k for k-means introduces systematic errors. OrthoRF, by producing clearly separated channels and leaving unused ones empty, naturally supports variable object counts.
> >
> > 5. **Zout vs Zfinal distinction.**
> >
> > We thank the reviewer for pointing out this inconsistency. To clarify: **($z_{\text{out}}$)** refers to the output of *any* layer in the network (i.e., the generic per-layer output), while **$z_{\text{final}}$** refers specifically to the output of the *last*/final layer. They are not interchangeable, and we will revise the manuscript to clearly distinguish the two terms.
> >
> > 6. **Typo in Table 2 (“thesh” → “thresh”).**
> > Thank you for identifying this typo, we will make sure to correct it.
> >
> > 7. **Capacity fairness.**
> > In Table 1, all models (AE, CAE, RF, OrthoRF) use the same architecture and number of parameters. The full architectural details—number of layers, kernel sizes, and channel configurations—are provided in Table 6 of the supplementary materials.
> >
> > 8. **Noise Robustness**:
> > In our SEM experiments, the added noise types and noise strengths are designed to reflect realistic distortions observed in SEM acquisitions. We have not yet evaluated more disruptive corruptions such as salt-and-pepper noise or a broader sweep over SNR levels. A systematic study of diverse noise types, and their interaction with synchrony-based binding, would be a valuable and natural direction for future work, and we will note this in the revision.
> >
> > 9. **Figure 2 and Figure 7 clarification.**
> >
> > In Figure 2, the choice of n=5 simply reflects the orientation dimensionality used in some of our 4Shapes experiments, which equals the number of objects (4) plus one extra component. This does not constrain OrthoRF to match the number of objects; in the supplementary material we include qualitative results for larger values (e.g., (n>5), Figures 9–12), demonstrating that the method also functions in overcomplete regimes.
> >
> > In Figure 7, the single-object images are used only for evaluation, specifically to measure performance on overlapping regions. They are **not** used during training, and **no label information is ever provided to the model**. These isolated-object images are included solely to mirror the evaluation protocol used for RF (Figure 6), where overlaps were excluded, and to allow a direct comparison under both evaluation settings.

---

### Official Review · Reviewer_WVWM · 2025-11-01

**Soundness:** 3
**Presentation:** 4
**Contribution:** 3
**Rating:** 8
**Confidence:** 3

**Summary:**

For the problem of forming multi-object representations, a previously proposed framework, "Rotating Features," can be improved by requiring the learned features to be orthogonal. This immediately improves the interpretability of the features, but surprisingly also improves performance in various tasks of synthetic scene understanding benchmarks.

**Strengths:**

* A straightforward method to add orthogonality constraints to a previously suggested loss.
* Improved performance (e.g., better handle occlusion), using a method which is much more scalable than other algorithms in the field (e.g., slot-based methods, synchrony-based methods).
* Improved interpretability, as the features are distinct rather than mixed (Figure 2 is impressive).

**Weaknesses:**

* The relative contribution of the two technical improvements ('Competitive binding in orientation space' and 'Orthogonality regularization') is not systematically analyzed, and it is unclear if both are needed. It seems the manuscript emphasizes the orthogonalization, but some of the heavy lifting is done by the improved 'gating' from the softmax.
* Inconsistent improvement over the previous method (as seen in Tables 1, 2, 3, 4).

**Questions:**

* Can you suggest in retrospect for which problems is the original RF method still superior (as seen in part in Tables 1, 2, 3, 4)?

---

> ### Author Response · Authors · 2025-11-25
>
> We appreciate your careful assessment of our work and the insightful comments. We address your points below. Please excuse the slight delay — additional experiments were required to provide a complete and well-supported response.
>
> 1. **Components contribution.**
> To clarify the individual contributions of competitive binding and orthogonality, we include the ablation table below.
> The ablation results demonstrate that the combination of competitive binding and orthogonality—not either component in isolation—is required to obtain the desired behaviour of OrthoRF. Only the full model achieves clean object separation (for n=5) and the strong performance reported in the paper (as seen in Fig. 3).
>
>
> | Variant                     | Notes                       | MSE ↓      | ARI ↑      | MBO ↑      |
> | --------------------------- | --------------------------- | ---------- | ---------- | ---------- |
> | RF                          | –                           | 0.0005     | 0.975      | 0.934      |
> | OrthoRF (no softmax, λ=0.1) | competition removed         | 0.0002     | 0.853      | 0.868      |
> | OrthoRF (softmax, λ=0)      | orthogonality removed       | 0.0034     | 0.628      | 0.688      |
> | **OrthoRF (full)**          | competition + orthogonality | **0.0002** | **0.9995** | **0.9887** |
>
>
> 2. **Inconsistent improvement over the previous method.**
> The four tables quantify different aspects of the model, and the improvements appear in different places depending on which component is being measured. Table 1 evaluates object discovery under mild occlusion. Here OrthoRF maintains similar performance to RF while gaining the additional ability to recover overlapping parts. Table 2 focuses on heavy occlusion and noise. In this more challenging setting, OrthoRF clearly surpasses RF.
> Tables 3 and 4 analyze the geometry of the learned representation. Table 3 shows that RF already exhibits orthogonal angles at the encoder output, even without explicit orthogonality. Table 4, however, demonstrates that OrthoRF achieves strong, consistent orthogonalization at the decoder output, with 90° inter-component separation and tight intra-cluster alignment, precisely the representational structure needed for reliable grouping. Taken together, these results show consistent improvements in the areas targeted by OrthoRF (representation geometry, robustness to occlusion/noise, overlap recovery), while maintaining competitive object discovery where RF already performs well. If any specific aspect seems unclear, we are happy to elaborate.
>
>
> 3. **RF advantages.**
> RF can still be advantageous in some settings. It reaches strong object-discovery performance with *distributed* representations that are easier to learn, which is why RF typically converges faster (e.g., ~100k steps vs. ~150k for OrthoRF). In contrast, OrthoRF optimizes for cleaner, object-specific channels, which is a harder objective but yields more interpretable features and occlusion recovery.

---

> > ### Comment · Reviewer_WVWM · 2025-11-26
> > **Response to authors**
> >
> > Thank you for your clarifications! I appreciate the effort in the ablation studies and will keep my current (high) score.

---

### Official Review · Reviewer_wSwc · 2025-11-01

**Soundness:** 3
**Presentation:** 3
**Contribution:** 3
**Rating:** 8
**Confidence:** 4

**Summary:**

This paper introduces Orthogonal Rotating Features (OrthoRF), an extension of the Rotating Features (RF) autoencoder framework for unsupervised object-centric learning (OCL). The original RF model, inspired by neural synchrony, uses vector-valued activations where magnitude encodes feature presence and orientation/phase encodes object affiliation, but often results in distributed representations requiring post-hoc clustering.

OrthoRF addresses this by imposing orthogonality as an inductive bias in the orientation space. It achieves this through two main architectural modifications:


1. Competitive Binding: A per-layer, centered softmax over orientation components to encourage "winner-take-most" specialization, mapping each object to a single, distinct component.

2. Orthogonality Regularization: An inner-product based $\mathcal{L}_{ortho}$ loss that penalizes the squared off-diagonal mass of the Gram matrix formed by the latent vectors, explicitly driving cross-component similarities toward zero, effectively enforcing a $90^{\circ}$ separation.

These modifications produce one-hot-like object encodings, which eliminates the need for post-hoc clustering and results in more interpretable representations. In experiments on the 4Shapes and synthetic SEM datasets, OrthoRF matches or outperforms current synchrony-based models on object discovery metrics (ARI-BG, MBO). Crucially, OrthoRF demonstrates superior performance on the shape completion task ($MBO_{i}^{OV}$), especially in overlapping/occluded regions, and shows a unique capability to recover occluded object parts in its intermediate representations—a strength not observed in prior slot-based or synchrony-based models.

**Strengths:**

* Originality and Significance: The core idea of enforcing orthogonality in the orientation space of RF is highly original and delivers a substantial improvement in the interpretability and performance of synchrony-based models. The result is a robust model that eliminates post-hoc clustering and yields one-hot-like encodings.

* Handling of Occlusion: OrthoRF exhibits a unique and powerful capability: the recovery of occluded object parts in its internal representations. This is demonstrated on the challenging, realistic SEM dataset and provides a crucial step toward more robust scene decomposition.

* Clarity and Efficacy: The method is conceptually clean (competitive binding + orthogonality loss) and empirically highly effective, with strong results across various conditions, noise, and out-of-distribution tests.

**Weaknesses:**

* Hyperparameter Sensitivity: The orthogonality loss is controlled by a weighting factor, $\lambda$. Table 1 shows that optimal performance varies across different settings of $n$ (orientation dimensionality), requiring a hyperparameter search for the specific dataset. For instance, $n=7$ prefers $\lambda=0.1$ while $n=5$ prefers $\lambda=0.8$. A more thorough sensitivity analysis or a proposed dynamic weighting strategy could improve the method's generalizability.

* Dependency on Final-Layer Output: The best-performing OrthoRF method (OrthoRF$^{\text{thresh}}$ on $\psi_{final}$) requires binarizing the intermediate map with a threshold (0.1) and no further post-processing. The dependence on a hand-tuned global threshold (0.1) for peak performance might limit its applicability compared to the fully unsupervised k-means approach used by RF and OrthoRF$^{\text{kmeans}}$. The reliance on an *intermediate* layer ($\psi_{final}$) rather than the final $z_{final}$ output for the $MBO_{i}^{OV}$ metric should be discussed as a limitation or characteristic.

* The authors claim training on noisy and testing on clean data degrades performance. I encourage authors to look at Extreme Image Transforms (EIT) [Crowder and Malik, 2022; Malik, Crowder and Mingolla, 2023, Biol Cybernetics]. They show that training on EITs helps with robustness on object detection tasks.

* The authors do not compare how their methods compare to other non-RF methods. I would encourage the authors to also compare their findings with methods like Stable and expressive recurrent vision models [Linsley et al., 2020, NeurIPS].

**Questions:**

1. The OrthoRF$^{\text{thresh}}$ results, particularly the striking $MBO_{i}^{OV}$ performance, rely on using the intermediate $\psi_{final}$ with a manually set global threshold of 0.1. Have the authors explored making this threshold adaptive, for instance, by linking it to the statistical properties of the activations (e.g., a multiple of the standard deviation) to maintain the method's unsupervised nature?

2. The paper uses a synthetic SEM dataset to show robustness to noise and occlusions. Could the authors elaborate on how they would apply the OrthoRF framework to a real, complex dataset, like CLEVR or MOVi-C, and what modifications might be necessary to the architecture or training protocol?

3. Given the effectiveness of orthogonality, have the authors considered applying the orthogonality loss constraint directly to slot-based models (like Slot Attention) in conjunction with their reconstruction or binding losses? This might provide further insight into the general utility of this inductive bias in OCL.

4. In relevant works, the authors have missed a few important papers to cite/compare their method with. When talking about ROOTS and SAVi, the work on PathTracker and InT [Linsley et al., 2021, NeurIPS] also disentangles objects in space. Similarly the extension of this work, FeatureTracker [Muzellec et al., 2025, ICLR] disentangles the changes in feature and colorspace.

5. I would highly encourage the authors to release the code publicly for further experimentation by the community.

6. The authors should also mention about the training details and hardware setup.

7. The citation for paper Rotating features for object discovery is repeated twice.

---

> ### Comment · Reviewer_wSwc · 2025-11-20
>
> Please let me know if there are any questions about the review. Thanks.

---

> > ### Author Response · Authors · 2025-11-25
> >
> > We are grateful for your time and the constructive suggestions outlined in your review. We address your points below. The brief delay was due to running new experiments to provide a thorough reply.
> >
> > **1. Sensitivity analysis / dynamic λ.**
> > We provide results across a wide range of λ values (see Table A in the response to Reviewer 1 with ID wHKK). OrthoRF is generally robust: different λ values give similar performance with only minor fluctuations. Table 1 in the paper reports the best λ for clarity, and small differences from the rebuttal reflect reruns. A λ of 0.1 works reliably in most settings; for large orientation spaces (e.g., n=20), performance is strong even with λ=0 due to the stabilizing architectural changes. While adaptive λ is an interesting future direction, our analysis shows that OrthoRF is not highly sensitive to this hyperparameter.
> >
> > **2. “Intermediate layer” clarification.**
> > We do *not* use an earlier network layer. The ψ representation used for occlusion completion is produced **in the final layer**. Each layer outputs both ψ and z. In the final layer, z reconstructs only visible parts, while ψ retains the full object structure. Thus ψ is an *intermediate representation inside the final layer*, not an earlier feature map.
> >
> > **3. Adaptive threshold.**
> > The ψ-thresholding step is rather insensitive. As shown in visualizations (to be added in the supplementary material), we first discard background channels (z-maps with very low variance), then threshold the remaining maps. A wide range of thresholds yields nearly identical binarizations because competitive binding produces a strong separation between active and inactive regions. The 0.1 value is simply a convenient default. An adaptive statistical threshold is a natural extension and will be mentioned.
> >
> > **4. EIT suggestion.**
> > We appreciate the pointer. Incorporating Extreme Image Transforms into synchrony-based models requires studying how such transformations interact with phase-based binding—a nontrivial and interesting direction for future work. We will note this in the paper.
> >
> > **5. Comparison to non-RF models.**
> > Thank you for the suggestions. Many proposed models (e.g., recurrent vision models) focus on temporal grouping, which differs from our static synchrony-based setting. Among non-RF baselines, **Slot Attention** is the most comparable for static OCL. We therefore include results on MNIST&Shape (CAE’s dataset), showing that synchrony-based methods remain competitive and that OrthoRF additionally recovers occluded object parts (visuals will be added to the supplementary material).
> >
> > | Model          | ARI+BG     | ARI–BG     |
> > | -------------- | ---------- | ---------- |
> > | CAE            | 0.783      | 0.971      |
> > | RF             | 0.6999     | 0.9720     |
> > | DBM            | 0.718      | 0.175      |
> > | Slot Attention | 0.047      | 0.089      |
> > | **OrthoRF**    | **0.6385** | **0.9927** |
> >
> > We will cite Linsley et al. , Muzellec et al. and related works in the revision.
> >
> > **6. Applying OrthoRF to CLEVR / MOVi-C.**
> > Applying OrthoRF to high-appearance-complexity datasets mainly requires a more expressive encoder and renderer. The binding mechanism and orthogonality principle remain fully applicable, and extending OrthoRF to such RGB domains is a promising next step. RF-style models can show sensitivity to color similarity, which makes this an interesting direction for future study.
> >
> > **7. Orthogonality in slot-based models.**
> > Orthogonality in OrthoRF is introduced to separate orientation components and reduce phase mixing. Slot Attention already achieves separation via iterative routing that forms discrete object slots, so imposing orthogonality on slots would not serve the same purpose and may interfere with their assignment process. Nonetheless, exploring orthogonality-based regularization in slot architectures is an interesting broader question.
> >
> > **8. Missing citations.**
> > Thank you — we will include PathTracker, InT, FeatureTracker, and related works.
> >
> > **9. Code release.**
> > As noted in the response to reviewer wHKK, code release is under internal review, and we hope to make it public. In the meantime, we will provide clear pseudocode and all configs.
> >
> > **10. Training details and hardware.**
> > Training hyperparameters are provided in the supplementary material (Tables 6–7), and hardware details are provided in the Implementation Details section. We will make sure that everything is described there.
> >
> > **11. Duplicate RF citation.**
> > Thank you — we will correct it.

---

> > > ### Comment · Reviewer_wSwc · 2025-11-27
> > >
> > > Thank you for the detailed rebuttal response. i look forward to seeing these in the main text/appendix of the paper for further understanding of the work to the end user.

---

### Official Review · Reviewer_wHKK · 2025-11-04

**Soundness:** 2
**Presentation:** 3
**Contribution:** 2
**Rating:** 4
**Confidence:** 2

**Summary:**

The paper introduces OrthoRF, a simple tweak to Rotating Features that makes each “orientation channel” specialize in a single object by adding a softmax competition and an orthogonality penalty. This reduces the usual phase-mixing and removes the need for post-hoc k-means, especially helping in overlapping/occluded regions where they can directly threshold an intermediate map to complete hidden parts. On synthetic 4Shapes and a custom SEM-style dataset, OrthoRF matches RF on standard object discovery and improves occlusion completion, with cleaner, near-orthogonal channels.

**Strengths:**

- The idea is intuitive: make channels compete and push them orthogonal so each object lands on its own axis. This yields cleaner, more interpretable features and avoids fragile clustering steps.
- The method is lightweight to implement (softmax + inner-product loss) and shows consistent angle statistics and tighter clusters. --- Qualitative plots match the quantitative gains, which is good to see. Table 2 shows substantial gains over RF in noisy conditions

**Weaknesses:**

- Most evidence is on binary synthetic images. It’s hard to judge real-world usefulness without RGB/textures or common OCL benchmarks.
- The headline occlusion gains rely on changing the decoding step. This makes it unclear how much of the win comes from the model vs. the decoding choice.
- The method still needs n set to at least the number of objects plus background. It’s not clear how it behaves when object count varies or is unknown at test time. The paper doesn't show what happens when n is too small (fewer components than objects), n is too large (many unused components) or object count changes at test time
- There’s no clear orthogonality ablation—it would help to see results when varying the orthogonality term, or comparing to alternative orthogonality formulations, to verify that this term is the main driver of the improvements.

**Questions:**

- Can you report both models with the same decoding strategy so we can separate model effects from decoding choices? A small table would clarify this quickly.
- Will you release code, configs, and a generator for the SEM dataset so others can reproduce the results?

---

> ### Author Response · Authors · 2025-11-25
>
> Thank you for your thoughtful review. We address your points below. Please excuse the slight delay caused by running additional experiments to provide a complete and well-supported response.
>
> **1. On the use of binary/synthetic datasets and lack of RGB/OCL benchmarks.**
> Our goal is to analyze which architectural ingredients improve synchrony-based object-centric models. Prior work shows that RF struggles on RGB images unless depth masks are provided (Table 10 & Fig. 6 in RF) and performs far below Slot-Attention models on Multi-dSprites and CLEVR (Table 8 in RF) due to color-based merging. For these reasons, we focus on domains where RF is well-motivated and practically effective, and we extend the evaluation to multi-object scenes with significant overlap, such as SEM images. The SEM-like dataset corresponds to a real high-value application domain in semiconductor metrology, where overlap recovery is essential and where RGB textures are not the main challenge. These controlled settings let us isolate the conceptual contribution of OrthoRF and study how orthogonality improves grouping and phase separation. Extending to complex RGB benchmarks is an important direction for future work.
>
>  **2. Decoding vs. model contribution.** We include an ablation using identical decoding for RF and all OrthoRF variants. The results (n=5) show that removing either the architectural competition or the orthogonality loss degrades OrthoRF, while their combination yields the strongest performance. Under the same decoding, OrthoRF also produces clean object channels and recovers occluded parts.
>
> **Ablation (same decoding for all models)**
>
> | Variant| Notes| MSE ↓| ARI ↑| MBO ↑|
> | --------------------------- | --------------------------- | ---------- | ---------- | ---------- |
> |RF| –| 0.0005| 0.975      | 0.934      |
> | OrthoRF (no softmax, λ=0.1) | architecture removed| 0.0002     | 0.853      | 0.868      |
> | OrthoRF (softmax, λ=0)      | loss removed | 0.0034     | 0.628      | 0.688      |
> | **OrthoRF (full)**          | competition + orthogonality | **0.0002** | **0.9995** | **0.9887** |
>
> **3. Behavior when (n) is too small, too large, or mismatched.** Test-time mismatch experiments are running and will be added to the final version of the paper in the supplement materials. Below, we provide extensive results for undercomplete and overcomplete regimes, and for varying λ.
>
>  **(a) Overcomplete (n).** For (n ∈ {5,6,7,8,9,20}) and λ ∈ {0,0.05,0.1,0.5,0.8}, OrthoRF remains stable even at (n=20) with extra channels staying unused. RF, however, degrades at n=20 (ARI=0.757, MBOi=0.774), likely because its distributed phase representation becomes unstable in higher-dimensional orientation spaces, leading to phase mixing.
>
> **Table A — Varying λ and n (OrthoRF)**
>
> **ARI**
>
> | λ    | 5     | 6     | 7     | 8     | 9     | 20   |
> | ---- | ----- | ----- | ----- | ----- | ----- | ---- |
> | 0.05 | .848  | .998  | .9996 | .994  | .996  | .997 |
> | 0.1  | .9997 | .999 | .9947 | .9994 | .9996 | .999 |
> | 0.5  | .9996 | .994  | .999  | .994  | .981  | .999 |
> | 0.8  | .9995 | .818  | .9992 | .998  | .9993 | .943 |
> | 0    | .628  | .600  | .968  | .790  | .965  | .998 |
>
> **MBOi**
>
> | λ    | 5    | 6    | 7    | 8    | 9    | 20   |
> | ---- | ---- | ---- | ---- | ---- | ---- | ---- |
> | 0.05 | .887 | .894 | .990 | .983 | .984 | .988 |
> | 0.1  | .985 | .876 | .986 | .988 | .989 | .988 |
> | 0.5  | .979 | .986 | .985 | .982 | .967 | .986 |
> | 0.8  | .989 | .861 | .987 | .988 | .993 | .893 |
> | 0    | .688 | .636 | .815 | .707 | .812 | .992 |
>
> **(b) Undercomplete (n)**
>
> When n < 4, both RF and OrthoRF must merge objects. OrthoRF degrades predictably and recovers perfect separation at n=4.
>
> **Table B — GRAY-4Shapes (OrthoRF vs RF)**
>
> | n | OrthoRF ARI | RF ARI | OrthoRF MBO | RF MBO |
> | - | --------------- | ---------- | --------------- | ---------- |
> |2|0.54|0.67|0.48|0.59|
> |3|0.65|0.94|0.68|0.86|
> |4|0.99|0.94|0.87|0.91|
>
> **4. Orthogonality ablation.** When n is small (5–6), λ=0 causes large drops in ARI/MBO, while even small λ restores strong performance by reducing phase mixing. For large (n), performance is high even with λ=0, but orthogonality still improves separation and consistency. This confirms that the orthogonality loss is a key driver of OrthoRF’s improvements, especially in the practically relevant low-(n) regime where the training is faster.
>
> **5. Code, configs, and SEM generator.** The full codebase is under internal review within our industry partner, and we aim to release it in the future. OrthoRF modifies RF with two transparent additions (an orthogonality loss and a binding-competition mechanism). We will provide clear pseudocode and all configuration files in the supplement material (as we have already done for the presented experiments). The SEM generator cannot be released due to confidentiality constraints, but the dataset characteristics and process are described in enough detail for others to reproduce similar SEM-style data.

---

### Author Response · Authors · 2025-12-03
**General Summary of Additions and Clarifications**

### **General Summary of Additions and Clarifications**

We provide this general comment to summarize the main updates and clarifications included in the rebuttal discussion. OrthoRF is a synchrony-based approach that reinterprets rotating features (RF) from distributed representations into distinct object representations, making it functionally closer to slot-based models. Through this mechanism, the model is able to reveal portions of occluded objects, as demonstrated across datasets in the main paper and appendix.
Below, we outline the key points raised by the reviewers and our corresponding clarifications.

1. **Unsupervised method:**
   The entire approach is fully unsupervised and does not require labels at any stage of training.

2. **Effect of the orthogonality coefficient λ:**
   When the number of slots/rotations n is large, performance remains consistently strong across a wide range of λ values.
   For smaller n, incorporating orthogonality regularization (λ > 0) becomes essential for achieving clean object separation.

3. **K-means is optional:**
   K-means clustering can be used for mask extraction but is not required.
   With OrthoRF, objects naturally occupy separate feature dimensions, making them easily identifiable without clustering.

4. **Thresholding procedure:**
   Thresholding is applied directly to the soft masks to obtain binary object masks for metric computation.
   The ψ-thresholding procedure is largely insensitive to the threshold value. As shown in Fig. 10 (appendix), we first discard background channels (feature maps with very low variance) and then apply a fixed threshold to the remaining maps.
   Due to competitive binding, active and inactive regions are well separated, so a wide range of thresholds produces nearly identical binarizations. We use 0.1 as a convenient default, though other values yield similarly strong results. An adaptive thresholding method is a natural direction for future work.

5. **Extended evaluation for different slot/rotation counts n:**
   We expand Table 1 to include scenarios where  n < number of objects and n \gg number of objects.
   Performance remains high even for very large n (e.g., n = 20 ), while using fewer slots than objects naturally reduces accuracy.

6. **Varying number of objects:**
   We include experiments on a publicly available dataset (Shapes) where each image contains a variable number of objects with randomly sampled shapes—meaning the same shape may appear multiple times in one image. OrthoRF outperforms competing models, including RF, in this setting (see Table 4).

7. **Additional architectural ablations:**
   We include new ablations isolating the effects of softmax + centering and the orthogonality loss.
   The results show that the combination of both components yields the strongest performance.

8. **Text revisions:**
   To accommodate the additional material requested by reviewers while remaining within the page limit, certain text sections were condensed—without altering their content or meaning.

We are grateful to the referees for their thorough evaluation and valuable suggestions. We feel that their feedback has significantly strengthened the manuscript.

---

### Meta-Review · Area_Chair_L8SE · 2026-01-13

**Summary:**

**Summary**

The paper presents OrthoRF, an enhancement of the Rotating Features (RF) framework for unsupervised object-centric learning. By introducing a softmax competition and an orthogonality penalty, OrthoRF allows each orientation channel to specialize in a single object, improving interpretability and eliminating the need for post-hoc clustering. The modifications lead to one-hot-like object encodings, enhancing performance in object discovery and occlusion completion tasks, particularly in overlapping regions. Experiments on synthetic datasets, 4Shapes and SEM, demonstrate that OrthoRF matches or surpasses existing models, showcasing its effectiveness in recovering occluded object parts and improving overall representation clarity.

**Strengths**

- **Originality and Significance**: The method introduces a novel approach by enforcing orthogonality in the orientation space of RF, significantly enhancing interpretability and performance in synchrony-based models, while eliminating the need for post-hoc clustering.

- **Handling of Occlusion**: OrthoRF effectively recovers occluded object parts in its internal representations, demonstrating its robustness in scene decomposition, particularly in challenging datasets.

- **Clarity and Efficacy**: The method is conceptually straightforward and empirically effective, showing strong performance across various conditions and providing clear, interpretable results that enhance the understanding of internal representation geometry.

**Weaknesses**

- **Limited Real-World Applicability**: Most evidence is based on binary synthetic images, lacking RGB/textures or common OCL benchmarks, making it difficult to assess real-world usefulness.

- **Unclear Contribution of Model Changes**: The reliance on changes in the decoding step for headline occlusion gains raises questions about the actual contributions of the model versus the decoding choice.

- **Fixed Capacity Issues**: The method requires a fixed number of components (n), which limits flexibility and generalization when the object count varies or is unknown at test time, potentially reintroducing rigidity associated with slot-based methods.

- **Lack of Comprehensive Comparisons and Analysis**: There is insufficient comparison with non-RF methods, and the relative contributions of the two technical improvements (competitive binding and orthogonality regularization) are not systematically analyzed, leading to ambiguity regarding their necessity and effectiveness.

**Decision**

The paper is recommended for acceptance, as it enforces orthogonality in RF orientation space and improves interpretability and performance in synchrony-based models without requiring post-hoc clustering. It effectively recovers occluded object parts, showcasing robustness in scene decomposition, especially in challenging datasets. The method is clear and empirically effective, demonstrating strong performance and providing interpretable results that enhance understanding of internal representation geometry, although there is a clear limitation to real-world applications.

**Reviewer Concerns:**

The author's feedback effectively addressed most concerns, enhancing the overall quality of the paper. Additionally, they clarified several ambiguous points, leading most reviewers to consider a slight increase in their scores.

**Reviewer Scores:**

The author's feedback effectively addressed most concerns, enhancing the overall quality of the paper. Additionally, they clarified several ambiguous points, leading most reviewers to consider a slight increase in their scores.

---

### Decision · Program_Chairs · 2026-01-26

Accept (Poster)